# MULTI-STEP PREFERENCE OPTIMIZATION VIA TWO-PLAYER MARKOV GAMES

## ABSTRACT

Reinforcement Learning from Human Feedback (RLHF) has been highly success-
ful in aligning large language models with human preferences. While prevalent
methods like DPO have demonstrated strong performance, they frame interactions
with the language model as a bandit problem, which limits their applicability in
real-world scenarios where multi-turn conversations are common. Additionally,
DPO relies on the Bradley-Terry model assumption, which does not adequately
capture the non-transitive nature of human preferences. In this paper, we address
these challenges by modeling the alignment problem as a two-player constant-sum
Markov game, where each player seeks to maximize their winning rate against the
other across all steps of the conversation. Our approach Multi-step Preference Op-
timization (MPO) is built upon the natural actor-critic framework (Peters & Schaal,
2008). We further develop OMPO based on the optimistic online gradient descent
algorithm (Rakhlin & Sridharan, 2013; Joulani et al., 2017). Theoretically, we
provide a rigorous analysis for both algorithms on convergence and show that
OMPO requires $\mathcal{O}(\epsilon^{-1})$ policy updates to converge to an $\epsilon$-approximate Nash equi-
librium. We also validate the effectiveness of our method through experiments on
the multi-turn conversations dataset in MT-bench-101.

## 1 INTRODUCTION

In recent years, the integration of large-language models (LLMs) (Brown et al., 2020; Achiam
et al., 2023; Team et al., 2023) into various applications has highlighted the need for advanced
preference alignment methods (Ziegler et al., 2019; Stiennon et al., 2020; Bai et al., 2022; Ouyang
et al., 2022; Rafailov et al., 2023). As models increasingly engage in complex decision making or
reasoning scenarios, e.g., GPT-4o and o1[1], the ability to align their outputs with user preferences has
received more attention. However, existing works on reinforcement learning from human feedback
(RLHF) focus mostly on one-step preference (Rafailov et al., 2023; Meng et al., 2024; Munos
et al., 2024; Azar et al., 2024; Wu et al., 2024; Zhang et al., 2024), which neglects indispensable
intermediate preferences within the answer and limits the model's alignment ability. For example,
in multi-round conversations, alignment must occur at each turn to meet user needs. Similarly, in
mathematical reasoning with chain-of-thought prompting, step-by-step validation is essential to
ensure accuracy in the final result. The reliance on final-output feedback in most existing RLHF
methods (Wang et al., 2023; Shani et al., 2024) neglects these intermediate steps, highlighting the
need for multi-step preference optimization to enhance alignment capabilities.

Meanwhile, earlier alignment methods e.g., DPO and its variants step-DPO (Lai et al., 2024; Lu
et al., 2024), typically model the pairwise preference by the Bradley-Terry model (Bradley & Terry,
1952), which assigns a score for each answer based on its preference. This assumption of the
model cannot capture the non-transitive preference, which is often observed in the averaged human
preferences from the population (Tversky, 1969; Gardner, 1970). While a recent line of work has
modeled the alignment process under the framework of general preference (Azar et al., 2024; Munos
et al., 2024; Wu et al., 2024; Rosset et al., 2024), and thus bypasses the BT model assumption, the
challenge of multi-step preference optimization remains underexplored.

In this paper, we first address this gap by formulating multi-step general preference optimization
within the framework of two-player Markov games (Shapley, 1953), where each player seeks to

---

[1]https://openai.com/o1

maximize their winning rate against the other across all steps of the conversation. Next, we introduce Multi-step Preference Optimization (MPO) drawing on insights from the natural actor-critic framework (Peters & Schaal, 2008). We further develop OMPO which leverages the optimistic online gradient descent algorithm and benefits from improved theoretical guarantees (Rakhlin & Sridharan, 2013; Joulani et al., 2017). Theoretically, we provide rigorous analysis for both algorithms on the convergence to Nash equilibrium. Empirically, we demonstrate the effectiveness of our approach through experiments on multi-turn conversation datasets, such as MT-bench-101. We firmly believe that our framework and approach can enhance the responsiveness of LLMs to user feedback.

Based on our discussions above, we summarize the contributions as follows:

- We formulate multi-step preference optimization as a two-player partially observable Markov game. Unlike Wang et al. (2023); Swamy et al. (2024); Shani et al. (2024) who focus on the preference feedback at the final state, we assume that the preference signal is received at each step. Such feedback allows the model to better identify which steps are correct or erroneous, potentially enhancing learning efficiency and accuracy.

- We propose Multi-step Preference Optimization (MPO) based on the natural actor-critic framework and Optimistic Multi-step Preference Optimization (OMPO), built upon the optimistic online gradient descent. Theoretically, we show that OMPO requires $\mathcal{O}(\epsilon^{-1})$ policy updates to converge to an $\epsilon$-approximate Nash equilibrium, compared to $\mathcal{O}(\epsilon^{-2})$ by the algorithms provided in Wang et al. (2023); Swamy et al. (2024); Shani et al. (2024). Our result cannot be trivially extended by Alacaoglu et al. (2022) due to the partially observable nature of Markov game. Interestingly, we bypass this difficulty by deriving our OMPO that parameterizes the game over occupancy measures.

- We provide practical implementations of both MPO and OMPO for LLM alignment. Numerical results show that the proposed methods achieve considerable improvement on multi-turn conversation datasets, such as MT-bench-101, compared to the multi-step variant of DPO.

The remaining part of this paper is organized as follows: Sec. 2 provides a comprehensive review and discussion of related work. In Sec. 3, we introduce the problem setting for the investigated multi-step RLHF. Sec. 4.1 and Sec. 4.2 introduce the proposed MPO and OMPO and provide a theoretical convergence analysis. Experimental results are present in Sec. 5. Conclusion, limitation, and future work are discussed in Sec. 6.

## 2 RELATED WORK

**RLHF under Bradley-Terry model.** Over the years, significant strides have been made towards developing RLHF algorithms from various perspectives under the Bradley-Terry model Bradley & Terry (1952). Earlier RLHF pipelines usually included supervised fine-tuning, learning a reward model, and reinforcement learning optimization with PPO (Ziegler et al., 2019; Stiennon et al., 2020; Bai et al., 2022; Ouyang et al., 2022). Due to the instability and scaling issues of such a pipeline, direct alignment methods such as DPO have been proposed to bypass the training of the reward model (Rafailov et al., 2023). Several follow-up methods, such as generalized preference optimization (GPO, Tang et al. 2024), use offline preference data to directly optimize pairwise preferences against a fixed opponent. A number of works have proposed reference-model-free method (Meng et al., 2024; Hong et al., 2024). In Meng et al. (2024), the impact of sequence length is mitigated by averaging the likelihood over the length of the sequence. In the multi-step scenario, several multi-step variants of DPO are introduced in the math reasoning task. Lu et al. (2024) initiate from an intermediate step in a correct reasoning process and increase the temperature to produce a flawed reasoning path leading to an incorrect answer. Meanwhile, Lai et al. (2024) leverage GPT-4 to detect the first incorrect step in a multi-step reasoning trajectory, then regenerate from that point to obtain the correct path. Together, these serve as the pair of samples for DPO.

**RLHF under general preferences.** The reward model in the Bradley-Terry model inherently implies transitivity in preferences. However, human preferences, especially the resulting averaged human preferences from populations, are usually nontransitive (Tversky, 1969; Gardner, 1970). To this end, Azar et al. (2024) outline a general framework for RLHF starting from general preference optimization and shows that DPO is a special case with the assumption of Bradley-Terry model. They further proposed IPO without such an assumption. Subsequently, Munos et al. (2024) try to solve the alignment of non-transitive general preferences using two-player nash learning in a bandit

setting. In their work, preferences are regularized through KL divergence to a reference policy, and they prove the convergence of the last iterative. In Swamy et al. (2024), multi-step alignment is considered while preference signals are only applied at the final step. Swamy et al. (2024) do not demonstrate the effectiveness of this framework in large language models. Wu et al. (2024) propose SPPO, studying bandit alignment under general preferences. They introduce a novel loss function that increases the log-likelihood of the selected response while decreasing that of the rejected response, in contrast to DPO. Rosset et al. (2024) start with the nash learning framework and propose Online DPO, which is an iterative version of DPO. Wang et al. (2023) provide theoretical analysis on multi-step RLHF under general preference while practice application is not explored. In Wang et al. (2023), the preference signal is given for the entire trajectory of an MDP while in this paper it is step-wise. Shani et al. (2024) study multi-step alignment under general preferences. However, unlike their approach where only preferences at the final states are considered, our work is built on a two-player Markov game which assumes that human preference is received at each step rather than only at the final step. Additionally, we leverage the optimistic online gradient descent to achieve a better convergence rate than Wang et al. (2023); Shani et al. (2024), and utilize Monte Carlo estimation with a small-scale pairwise reward model, avoiding the need for an additional function approximator for the critic network.

**Two-player Markov game & optimistic online gradient descent.** Two-player Markov games have been widely studied since the seminal work (Shapley, 1953). Particularly relevant to our work is the research line on policy gradient algorithms for two-player Markov games such as Daskalakis et al. (2020); Wei et al. (2021); Alacaoglu et al. (2022). Our `OMPO` is strictly related to the idea of optimistic online gradient descent (Popov, 1980; Chiang et al., 2012; Rakhlin & Sridharan, 2013) originally proposed in online learning to achieve small regret in case of slow varying loss sequences. Our update that uses only one projection per update was proposed in Joulani et al. (2017). The name of our method is due to a similar algorithm introduced in the context of variational inequalities by Malitsky & Tam (2020).

## 3 MULTI-STEP RLHF AS TWO-PLAYER MARKOV GAMES

### 3.1 NOTATION

We define the prompt to the language model as $x$ and the answer from the language model as $a$. For a multi-turn conversation with turn $H$, the prompts and the answers are denoted by $x_h$ and $a_h, \forall h \in [H]$. The concatenation of a prompt $x$ and an answer $a$ is denoted by $[x, a]$ and can be generalized to the concatenation of multiple prompts and answers, e.g., $[x_1, a_1, \ldots, x_H, a_H]$. For any two sentences, e.g., $[x, a]$ and $[x', a']$, we define a preference oracle as $o([x, a] \succ [x', a']) \in \{0, 1\}$, which can provide preference feedback with 0-1 scores, where 1 means the conversation $[x, a]$ is preferred and 0 otherwise. We denote $\mathbb{P}([x, a] \succ [x', a']) = \mathbb{E}[o([x, a] \succ [x', a'])]$ as the probability that the conversation $[x, a]$ is preferred over $[x', a']$. Moreover, we have $\mathbb{P}([x, a] \succ [x', a']) = 1 - \mathbb{P}([x', a'] \succ [x, a])$. An autoregressive language model is denoted by $\pi(a|x)$ which receives input $x$ and generates answer $a$. We denote the KL divergence of two probability distributions $p$ and $q$ by $D(p\|q)$. The Bregman Divergences between two points are denoted by $\mathbb{D}(p\|q)$. The sigmoid function is defined by $\sigma(z) := \frac{1}{1+e^{-z}}$. Detailed definitions for the notations are summarized in Appx. A.

### 3.2 PROBLEM FORMULATION OF MULTI-STEP RLHF

In this section, we introduce the problem setting for multi-step RLHF and we defer the preliminaries on single-step RLHF to Appx. B. Specifically, we can cast the multi-step alignment process as a finite-horizon Markov Decision Process (MDP). We define $s_h = [x_1, a_1, \ldots, x_{h-1}, a_{h-1}, x_h]$ as the state at $h > 1$. We define the action $a_h$ as the answer given $s_h$. Particularly, we have $s_1 = x_1$. The prompt in the next state is sampled under the transition $x_{h+1} \sim f(\cdot|s_h, a_h)$, which is equivalent to $s_{h+1} \sim f(\cdot|s_h, a_h)$. The equivalence comes from the fact $s_{h+1} = [s_h, a_h, x_{h+1}]$ by using the concatenation operator between sentences. The terminal state is $s_{H+1}$. Our setting covers a number of alignment problems, and we list some examples below.

**Example 1** (Single-step alignment). *In single-step alignment, a language model receives one prompt and outputs one answer. Our framework covers the single-step alignment by dissecting the answer into single tokens. Specifically, we set $x_1$ as the prompt, $x_2, \ldots, x_{H+1}$ as empty sentences,*

*and the answer $a_h$ at each turn consists of only one token. Then the horizon $H$ is the number of tokens in the answer. The transition between each state is deterministic.*

**Example 2** (Chain-of-thought reasoning alignment). *In the chain-of-thought reasoning, the horizon $H$ denotes the number of reasoning steps, where $x_1$ is the initial prompt and $x_2, \ldots, x_{H+1}$ are empty. Each $a_h$ corresponds to a reasoning step. The transition between each state is deterministic.*

**Example 3** (Mutli-turn conversation alignment). *In multi-turn conversation, the horizon $H$ denotes the total number of turns in the conversation. In the $h$-th turn, $x_h$ is the prompt, and $a_h$ is the answer. The prompt in the terminal state, $x_{H+1}$, is an empty sentence. The transition between each state can be deterministic or stochastic.*

Next, we define the pair-wise reward function of two state-action pairs as the preference of two trajectories:

$$r(s_h, a_h, s_h', a_h') = \mathbb{P}([s_h, a_h] \succ [s_h', a_h']) .$$

Upon this point, we can define the MDP as a tuple $\mathcal{M} = (\mathcal{S}, \mathcal{A}, f, r, \nu_1, H)$, where $\mathcal{S}$ is the state space, $\mathcal{A}$ is the action space, $H$ is the horizon (total steps), the initial state distribution $\nu_1$ is a distribution over the initial prompt $x_1$. Note that in a two-player game environment, each state in $\mathcal{S}$ is a pair of $s_h$ and $s_h'$ generated by two policies. Our goal is to identify the Nash equilibrium (or von Neumann winner) of the following two-player constant-sum Markov game:

$$(\pi^*, \pi^*) = \arg\max_\pi \min_{\pi'} \mathbb{E}_{s_1 \sim \nu_1, s_h, a_h, s_h', a_h'} \Big[ \sum_{h=1}^{H} r(s_h, a_h, s_h', a_h') \Big], \qquad \text{(Game)}$$

where $s_1 = s_1' = x_1, a_h \sim \pi(\cdot|s_h), a_h' \sim \pi'(\cdot|s_h'), s_h \sim f(\cdot|s_{h-1}, a_{h-1}), s_h' \sim f(\cdot|s_{h-1}', a_{h-1}')$.

Here we make a few remarks on the benefit of incorporating human preferences at each step. More detail on the motivation can be found at Appx. G.

**Remark 1.** *If two conversations of $H$ turns, $s_{H+1}$ and $s_{H+1}'$, are globally similar but differ in the early turns (e.g., $s_2$ are better than $s_2'$), more credit should be assigned to $s_{H+1}$, encouraging the model to align with it. This follows the principle that humans typically master simpler and earlier tasks before progressing to more complex ones.*

**Remark 2.** *From a practical standpoint, including per-step preference data generates a richer dataset for training, helping the model learn which reasoning steps are correct or wrong. This incremental feedback can enhance overall performance by reinforcing the importance of foundational steps in reasoning.*

Next, we present some additional notation. We define the *pair-wise* value function as follows

$$V_h^{\pi, \pi'}(s, s') = \mathbb{E}\Big[ \sum_{\hat{h}=h}^{H} r(s_{\hat{h}}, a_{\hat{h}}, s_{\hat{h}}', a_{\hat{h}}')|s_h = s, s_h' = s' \Big] ,$$

where $a_{\hat{h}} \sim \pi_{\hat{h}}(\cdot|s_{\hat{h}}), a_{\hat{h}}' \sim \pi_{\hat{h}}'(\cdot|s_{\hat{h}}'), s_{\hat{h}+1} \sim f(\cdot|s_{\hat{h}}, a_{\hat{h}})$, and $s_{\hat{h}+1}' \sim f(\cdot|s_{\hat{h}}', a_{\hat{h}}')$. We will often denote $V_1^{\pi, \pi'}$ omitting the subscript, i.e., as $V^{\pi, \pi'}$. Moreover, notice that we consider potentially non stationary policies, i.e. they are indexed by $h$. We denote by $\pi$ the non stationary policy and by $\pi_h$ the distribution over actions at step $h$ corresponding to the non stationary policy $\pi$.

We define the *pair-wise* Q-function as follows:

$$Q_h^{\pi, \pi'}(s, a, s', a') = r(s, a, s', a') + \mathbb{E}\Big[ \sum_{\hat{h}=h+1}^{H} r(s_{\hat{h}}, a_{\hat{h}}, s_{\hat{h}}', a_{\hat{h}}') \Big] ,$$

where $s_{\hat{h}+1} \sim f(\cdot|s_{\hat{h}}, a_{\hat{h}})$ and $s_{\hat{h}+1}' \sim f(\cdot|s_{\hat{h}}', a_{\hat{h}}')$.

**Lemma 1.** *(Adapted from Puterman (1994)) The pair-wise value function and pair-wise Q-value function satisfy the following Bellman equation for all $h \in [H]$.*

$$Q_h^{\pi, \pi'}(s, a, s', a') = r(s, a, s', a') + \mathbb{E}_{\hat{s} \sim f(\cdot|s, a), \bar{s} \sim f(\cdot|s', a')}[V_{h+1}^{\pi, \pi'}(\hat{s}, \bar{s})] .$$

$$V_h^{\pi, \pi'}(s, s') = \mathbb{E}_{a \sim \pi_h(\cdot|s), a' \sim \pi_h'(\cdot|s')} Q_h^{\pi, \pi'}(s, a, s', a') .$$

By Lemma 1, we can rewrite Game as follows:

$$(\pi^*, \pi^*) = \arg\max_{\pi} \min_{\pi'} \mathbb{E}\Big[ \sum_{h=1}^{H} r(s_h, a_h, s'_h, a'_h) \Big] = \arg\max_{\pi} \min_{\pi'} \mathbb{E}_{s_1 \sim \nu_1} V^{\pi, \pi'}(s_1, s_1). \quad (1)$$

Given the above notation, we can formalize our objective. We look for a policy $\pi$ satisfying the following definition of approximate equilibrium.

**Definition 1** ($\epsilon$-**approximate Nash equilibrium**). *A policy $\pi$ is said to be an approximate Nash equilibrium if it holds that*

$$\langle \nu_1, V^{\pi, \pi} \rangle - \min_{\bar{\pi} \in \Pi} \langle \nu_1, V^{\pi, \bar{\pi}} \rangle \leq \epsilon,$$

*and*

$$\max_{\bar{\pi} \in \Pi} \langle \nu_1, V^{\bar{\pi}, \pi} \rangle - \langle \nu_1, V^{\pi, \pi} \rangle \leq \epsilon.$$

**Definition 2** (Occupancy measures). *Given the policy $\pi$, the occupancy measure of $\pi$, is defined at stage $h$ as $d_h^\pi(s, a) = \Pr(s_h = s, a_h = a)$ where $s_1 = x_1 \sim \nu_1, a_h \sim \pi_h(\cdot|s_h), s_h \sim f(\cdot|s_{h-1}, a_{h-1})$. We also define $d_h^\pi(s, a)|s_1 = \Pr(s_h = s, a_h = a|s_1 = s_1)$. In addition, given the policies $\pi, \bar{\pi}$, the occupancy measure of $(\pi, \bar{\pi})$ at stage $h$ is defined as $d_h^{\pi, \bar{\pi}}(s, a, s', a') = \Pr(s_h = s, a_h = a, s'_h = s', a'_h = a')$, where $s_1 = s'_1 = x_1 \sim \nu_1, a_h \sim \pi(\cdot|s_h), a'_h \sim \pi'(\cdot|s'_h), s_h \sim f(\cdot|s_{h-1}, a_{h-1}),$ and $s'_h \sim f(\cdot|s'_{h-1}, a'_{h-1})$.*

**Remark**: The value function at the initial state can be represented as an inner product between the reward function and the occupancy measure, i.e., $V^{\pi, \bar{\pi}} = \sum_{h=1}^{H} \langle r, d_h^{\pi, \bar{\pi}} \rangle$. Given the structure of the game where the sequences of sentences and answers are generated independently by the two agents given the initial state $s_1$, the occupancy measure at each step can be factorized as the product of the two agents occupancy measures given $s_1$. In particular, we have $d_h^{\pi, \bar{\pi}}(s, a, s', a')|s_1 = d_h^\pi(s, a)|s_1 \cdot d_h^{\bar{\pi}}(s', a')|s_1$ for all $h, s, a, s', a'$.

## 4 METHOD

We first develop our method Multi-Step Preference Optimization (MPO) based on the natural actor-critical framework (Peters & Schaal, 2008; Alacaoglu et al., 2022) in Sec. 4.1. Next, we introduce Optimistic Multi-Step Preference Optimization, dubbed OMPO, in Sec. 4.2. The framework is inspired by the idea of optimism used in online learning and in min-max optimization with improved theoretical guarantees (Popov, 1980; Chiang et al., 2012; Rakhlin & Sridharan, 2013).

### 4.1 MPO WITH NATURAL ACTOR-CRITIC

This section presents our first method to find an approximate solution to Game. In order to find an $\epsilon$-approximate Nash equilibrium, the MPO method builds upon the next lemma which decomposes the difference of two value functions to the $Q$ function at each step. The lemma 2 is the extension of Kakade & Langford (2002) to the multi-agent setting where the dynamics are controlled independently by each player but the reward depends on the joint-state action tuple.

**Lemma 2** (Value difference lemma (Adapted from Kakade & Langford (2002))). *For a finite horizon MDP with initial distribution $\nu_1$ it holds that:*

$$\left\langle \nu_1, V^{\pi, \bar{\pi}} - V^{\pi', \bar{\pi}} \right\rangle = \mathbb{E}_{s_1 \sim \nu_1} \sum_{h=1}^{H} \mathbb{E}_{s \sim d_h^\pi|s_1} \left[ \left\langle \mathbb{E}_{s', a' \sim d_h^{\bar{\pi}}|s_1} Q_h^{\pi', \bar{\pi}}(s, \cdot, s', a'), \pi_h(\cdot|s, s_1) - \pi'_h(\cdot|s, s_1) \right\rangle \right].$$

The proof can be found at Appx. D.2. In our setting, the initial state $s_1$ is a deterministic function of the state $s$ so we can remove $s_1$ from the conditioning in the policy[2]. To highlight this fact we

---

[2]This is motivated by practical LLM training, where system prompts such as "user" and "assistant" are inserted before every $x_h$ and $a_h$, respectively. As a result, one can infer a unique $s_1$ for every $s$. The conditioning of the policy on the initial state might appear unusual at the first glance but it is in fact common in the setting of Contextual MDPs (see for example Levy et al. (2023)). Indeed, the initial state $s_1$ could be interpreted as a context and we optimize over policies that depend on both the initial context and the current state.

---

**Algorithm 1** MPO (Theory Version)

---

**input**: reference policy $\pi^1$, preference oracle $\mathbb{P}$, learning rate $\beta = \sqrt{\frac{\log \pi^{-1}}{TH^2}}$, total iteration $T$

**for** $t = 1, 2, \ldots, T$ **do**

$$\pi_h^{t+1}(a|s) \propto \pi_h^t(a|s) \exp\left[\beta \mathbb{E}_{s', a' \sim d_h^{\pi^t}|s_1(s)} Q_h^{\pi^t, \pi^t}(s, a, s', a')\right] \quad \forall h \in [H], \quad \forall s, a.$$

**end for**

**output**: $\bar{\pi}^T$ ( such that $d_h^{\bar{\pi}^T} = \frac{1}{T}\sum_{t=1}^T d_h^{\pi^t}, \ \forall h \in [H]$. ).

---

**Algorithm 2** MPO (Practical version)

---

**input**: reference policy $\pi^1$, preference oracle $\mathbb{P}$, learning rate $\beta$, number of generated samples $K$, horizon $H$, total iteration $T$.

**for** $t = 1, 2, \ldots, T$ **do**

    Generates response by sampling $s_1^1 \sim \nu_1$ and $a_h^1 \sim \pi^t(\cdot|s_h^1)$ for $h \in [H]$.

    Clear the dataset buffer $\mathcal{D}_t$.

    **for** $h = 1, 2, \ldots, H$ **do**

        Set $s_h^K =, \ldots, = s_h^2 = s_h^1$.

        Generate $K - 1$ conversations by sampling $a_{\hat{h}}^{2:K} \sim \pi^t(\cdot|s_{\hat{h}}^{2:K})$ for $\hat{h} \in [h, H]$.

        Estimate $\mathbb{E}_{a_h^{k'}} Q^{\pi^t, \pi^t}(s_h^1, a_h^k, s_h^1, a_h^{k'}), \forall k, k' \in [K]$ via Eq. (5) with query to $\mathbb{P}$.

        Form the data pair $\{(s_h^1, a_h^k, \mathbb{E}_{a_h^{k'}} Q^{\pi^t, \pi^t}(s_h^1, a_h^k, s_h^1, a_h^{k'})\}_{k \in [K]}$ , add to $\mathcal{D}_t$ .

    **end for**

    Optimize $\pi_{t+1}$ over $\mathcal{D}_t$ according to

$$\pi^{t+1} \leftarrow \arg\min_\pi \mathbb{E}\left(\log\left(\frac{\pi(a_h^k|s_h^1)}{\pi^t(a_h^k|s_h^1)}\right) - \beta\left(\mathbb{E}_{a_h^{k'}} Q^{\pi^t, \pi^t}(s_h^1, a_h^k, s_h^1, a_h^{k'}) - \frac{H - h + 1}{2}\right)\right)^2.$$

**end for**

**output**: $\pi^{T+1}$

---

denote as $s_1(s)$ the only initial state that can lead to $s$. By setting $\pi' = \bar{\pi} = \pi^t$ in Lemma 2 and $\pi = \pi^\star$ and summing from $t = 1$ to $T$ we obtain:

$$\sum_{t=1}^T \left\langle \nu_1, V^{\pi^\star, \pi^t} - V^{\pi^t, \pi^t} \right\rangle = \mathbb{E}_{s_1 \sim \nu_1} \sum_{h=1}^H \sum_{t=1}^T \mathbb{E}_{s \sim d_h^{\pi^\star}|s_1}\left[\left\langle \mathbb{E}_{s', a' \sim d_h^{\pi^t}|s_1} Q_h^{\pi^t, \pi^t}(s, \cdot, s', a'), \pi_h^\star(\cdot|s) - \pi_h^t(\cdot|s)\right\rangle\right].$$

Since the sum over $t$ commutes with the expectation, we see that we can decompose the global regret $\sum_{t=1}^T \left\langle \nu_1, V^{\pi^\star, \pi^t} - V^{\pi^t, \pi^t}\right\rangle$ into a weighted sum of local regrets at each stage $h \in [H]$, i.e.,

$\mathbb{E}_{s \sim d_h^{\pi^\star}|s_1}\left[\sum_{t=1}^T \left\langle \mathbb{E}_{s', a' \sim d_h^{\pi^t}|s_1} Q_h^{\pi^t, \pi^t}(s, \cdot, s', a'), \pi_h^\star(\cdot|s) - \pi_h^t(\cdot|s)\right\rangle\right]$. Therefore, we can control the global regret implementing at each state online mirror descent updates (Warmuth et al. 1997, Orabona 2023, Chapter 6, Cesa-Bianchi & Lugosi 2006), i.e., implementing the following update:

$$\pi_h^{t+1}(\cdot|s) = \arg\max_\pi \langle \pi(\cdot|s), \mathbb{E}_{s', a' \sim d_h^{\pi^t}|s_1(s)} Q_h^{\pi^t, \pi^t}(s, \cdot, s', a')\rangle - \beta D(\pi(\cdot|s)||\pi_h^t(\cdot|s)),$$

where $\beta$ is a learning rate. The solution has the following form:

$$\pi_h^{t+1}(a|s) \propto \pi_h^t(a|s) \exp\{\beta \mathbb{E}_{s', a' \sim d_h^{\pi^t}|s_1(s)} Q_h^{\pi^t, \pi^t}(s, a, s', a')\}, \tag{2}$$

which corresponds to natural actor-critic (Peters & Schaal, 2008) that utilizes a softmax-based method for updating policies. The number of policy updates needed by the ideal version of MPO (see Alg. 1) can be bounded as follows and the proof can be found at Appx. D.3.

**Theorem 4.** *Consider Algorithm 1 and assume that the reference policy is uniformly lower bounded by $\underline{\pi}$, then there exists a policy $\bar{\pi}^T$ such that $d_h^{\bar{\pi}^T} = \frac{1}{T}\sum_{t=1}^T d_h^{\pi^t}, \forall h \in [H]$, and it holds that for $T = \frac{16H^4 \log \underline{\pi}^{-1}}{\epsilon^2}$ the policy pair $(\bar{\pi}^T, \bar{\pi}^T)$ is an $\epsilon$-approximate Nash equilibrium. Therefore, Algorithm 1 outputs an $\epsilon$-approximate Nash equilibrium after $\frac{16H^4 \log \underline{\pi}^{-1}}{\epsilon^2}$ policy updates.*

---

**Algorithm 3** OMPO (Theory Version)

---

**input**: occupancy measure of reference policy $\pi^1$ denoted as $d^1$, preference oracle $\mathbb{P}$ (i.e. reward function $r$), learning rate $\beta$, Bregman divergence $\mathbb{D}$, iteration $T$
**for** $t = 1, 2, \ldots, T$ **do**

$$d_h^{t+1} = \arg\max_{d \in \mathcal{F}_{s_1}} \beta \left\langle d, 2\mathbb{E}_{s',a' \sim d_h^t} r(\cdot, \cdot, s', a') - \mathbb{E}_{s',a' \sim d_h^{t-1}} r(\cdot, \cdot, s', a') \right\rangle - \mathbb{D}(d, d_h^t) \quad \forall h \in [H] \ \forall s_1.$$

**end for**
$\pi_h^{\text{out}}(a|s) = \frac{\bar{d}_h(s,a|s_1)}{\sum_a \bar{d}_h(s,a|s_1)}$ with $\bar{d}_h = T^{-1} \sum_{t=1}^T d_h^t$ for all $h \in [H]$ for the unique $s_1$ from which $s$ is reachable.
**Output :** $\pi^{\text{out}}$

---

**Remark 3.** *The above result generalizes the $\mathcal{O}(H^2 \epsilon^{-2})$ bound on the policy updates proven in Swamy et al. (2024) in the setting of terminal-only reward. The additional $H^2$ factor in our theorem is due to considering rewards that are not terminal-only. In Theorem 5 we show that Algorithm 3 improves the number of policy updates needed to converge to an $\epsilon$-approximate Nash equilibrium to $\mathcal{O}(H \epsilon^{-1})$.*

**Practical relaxations.** For the above theorem, MPO requires the access of the $Q$ function, which is unknown. Next, we are going to develop a practical algorithm to efficiently estimate the $Q$ function and implement Eq. (2). Equivalently, Eq. (2) can be written as

$$\pi_h^{t+1}(a|s) = \frac{\pi_h^t(a|s) \exp\{\beta \mathbb{E}_{s',a' \sim d_h^t | s_1(s)} Q_h^{\pi^t, \pi^t}(s, a, s', a')\}}{Z_h^t(s)}, \tag{3}$$

where $Z_h^t(s)$ is the partition function. Next, we express Eq. (3) as follows:

$$\log \frac{\pi_h^{t+1}(a|s)}{\pi_h^t(a|s)} = \beta \mathbb{E}_{s',a' \sim d_h^t | s_1(s)} Q_h^{\pi^t, \pi^t}(s, a, s', a') - \log Z_h^t(s). \tag{4}$$

Next, we approximate Eq. (4) with an approximate solution of the following optimization program

$$\pi^{t+1} = \arg\min_{\pi} \sum_{h=1}^H \mathbb{E}_{\substack{s_1 \sim \nu_1 \\ (s_h, a_h) \sim d_h^{\pi^t} | s_1}} \left[ \log \frac{\pi(a_h|s_h)}{\pi_h^t(a_h|s_h)} - (\mathbb{E}_{s',a' \sim d_h^{\pi^t} | s_1} Q_h^{\pi^t, \pi^t}(s_h, a_h, s', a') - \log Z_h^t(s_h)) \right]^2.$$

Unfortunately, solving the above minimization exactly is out of hope. The first difficulty is the efficient estimation of $\mathbb{E}_{s',a' \sim d_h^t | s_1} Q_h^{\pi^t, \pi^t}(s_h, a_h, s', a')$. In particular, since $s'$ and $s$ are sampled from the same distribution, we will sample $a'$ from the state $s_h$ and use the Monte Carlo estimator:

$$\mathbb{E}_{a' \sim \pi^t(\cdot|s_h)} Q_h^{\pi^t, \pi^t}(s_h, a_h, s_h, a') \approx \frac{1}{K} \sum_{k=1}^K \sum_{\hat{h}=h}^H \mathbb{P}([s_{\hat{h},k}, a_{\hat{h},k}], [s'_{\hat{h},k}, a'_{\hat{h},k}]), \tag{5}$$

where the sequences $\left\{ (s_{\hat{h},k}, a_{\hat{h},k}, s'_{\hat{h},k}, a'_{\hat{h},k}) \right\}_{\hat{h}=h}^H$ for $k \in [K]$ are generated by rollouts of the policies pair $(\pi^t, \pi^t)$. The second difficulty is $Z_h^t(s)$, which is difficult to compute for large action spaces. In all states $s$, we replace $\log Z_h^t(s)$ with $\beta \frac{H-h+1}{2}$.

**Remark 4.** *The heuristics is motivated by the next observation. If the preference between $a_h$ and $a_h'$ in Eq. (5) results in a tie, then with such $\log Z_h^t(s)$, the solution of Eq. (5) is $\pi^{t+1} = \pi^t$, leaving the model unchanged.*

In summary, we provide a practical version of MPO in Alg. 2. In practice, we used a stationary policy that we find to be sufficient to obtain convincing results.

### 4.2 OPTIMISTIC MPO: OMPO

In this section, we propose an alternative algorithm based on the optimistic gradient descent method [3] by reformulating the optimization problem over occupancy measures. Here, we show that opti-

---

[3]The same update we use can also be seen as the Forward-Reflected-Backward (FoRB) update proposed in Malitsky & Tam (2020) for variational inequalities. This point of view is taken by Alacaoglu et al. (2022) to solve zero-sum Markov game.

---

**Algorithm 4** OMPO (Practical version)

---

**input**: reference policy $\pi^1$, preference oracle $\mathbb{P}$, learning rate $\beta$, number of generated samples $K$, horizon $H$, total iteration $T$, tunable bias term $\tau$.

**for** $t = 1, 2, \ldots, T$ **do**

    Generates response by sampling $s_1^1 \sim \nu_1$ and $a_h^1 \sim \pi^t(\cdot|s_h^1)$ for $h \in [H]$.

    Clear the dataset buffer $\mathcal{D}_t$.

    **for** $h = 1, 2, \ldots, H$ **do**

        Set $s_h^K =, \ldots, = s_h^2 = s_h^1$.

        Generate $K - 1$ conversations by sampling $a_{\hat{h}}^{2:K} \sim \pi^t(\cdot|s_{\hat{h}}^{2:K})$ for $\hat{h} \in [h, H]$.

        Estimate $\mathbb{E}_{a_h^{k'}} Q^{\pi^t, \pi^t}(s_h^1, a_h^k, s_h^1, a_h^{k'}) \forall k, k' \in [K]$ via Eq. (5).

        **if** $t > 1$ **then**

            Estimate $\mathbb{E}_{a_h^{k'}} Q^{\pi^t, \pi^{t-1}}(s_h^1, a_h^k, s_h^1, a_h^{k'}) \quad \forall k, k' \in [K]$ via Eq. (5).

            Add $\{(s_h^1, a_h^k, \mathbb{E}_{a_h^{k'}} Q^{\pi^t, \pi^t}(s_h^1, a_h^k, s_h^1, a_h^{k'}), \mathbb{E}_{a_h^{k'}} Q^{\pi^t, \pi^{t-1}}(s_h^1, a_h^k, s_h^1, a_h^{k'})\}_{k \in [K]}$ into $\mathcal{D}_t$.

        **else**

            Add $\{(s_h^1, a_h^k, \mathbb{E}_{a_h^{k'}} Q^{\pi^t, \pi^t}(s_h^1, a_h^k, s_h^1, a_h^{k'})\}$ into $\mathcal{D}_t$.

        **end if**

    **end for**

    **if** $t > 1$ **then**

        Optimize $\pi_{t+1}$ over $\mathcal{D}_t$ according to

$$\pi^{t+1} \leftarrow \arg\min_{\pi} \mathbb{E}\left( \log\left( \frac{\pi(a_h^k|s_h^1)}{\pi^t(a_h^k|s_h^1)} \right) - \beta\left( 2\mathbb{E}_{a_h^{k'}} Q^{\pi^t, \pi^t}(s_h^1, a_h^k, s_h^1, a_h^{k'}) - \mathbb{E}_{a_h^{k'}} Q^{\pi^t, \pi^{t-1}}(s_h^1, a_h^k, s_h^1, a_h^{k'}) - \tau \right) \right)^2.$$

    **else**

        Optimize $\pi_{t+1}$ over $\mathcal{D}_t$ according to

$$\pi^{t+1} \leftarrow \arg\min_{\pi} \mathbb{E}\left( \log\left( \frac{\pi(a_h^k|s_h^1)}{\pi^t(a_h^k|s_h^1)} \right) - \beta\left( \mathbb{E}_{a_h^{k'}} Q^{\pi^t, \pi^t}(s_h^1, a_h^k, s_h^1, a_h^{k'}) - \frac{H - h + 1}{2} \right) \right)^2.$$

    **end if**

**end for**

**output**: $\pi^{T+1}$

---

mistic online mirror descent with one projection (Joulani et al., 2017) with an appropriately chosen regularizer can be used to solve approximately the following program which corresponds to Game lifted to the space of conditional occupancy measures.

$$(d^\star, d^\star) = \arg\max_{d \in \tilde{\mathcal{F}}} \min_{d' \in \tilde{\mathcal{F}}} \mathbb{E}_{s_1 \sim \nu_1} \sum_{h=1}^{H} \sum_{s, a, s', a'} d_h(s, a|s_1) r(s, a, s', a') d_h'(s', a'|s_1),$$

where $\tilde{\mathcal{F}}$ is the product set of the Bellman flow constraints for a particular initial state, i.e. $\tilde{\mathcal{F}} = \times_{s_1 \in \text{supp}(\nu_1)} \mathcal{F}_{s_1}$. We also introduced the Bellman flow constraints for a specific initial state $\mathcal{F}_{s_1} = \left\{ d = (d_1, \ldots, d_H) : \sum_a d_{h+1}(s, a) = \sum_{s', a'} f(s|s', a') d_h(s', a'), d_1(s) = \mathbb{1}\{s = s_1\} \right\}$. The policy pair $(\pi^\star, \pi^\star)$ solution of Game can be retrieved from the occupancy measure pair $(d^\star, d^\star)$ as $\pi^\star(a|s) = \frac{d^\star(s, a|s_1)}{\sum_a d^\star(s, a|s_1)}$. Our idea is to apply the optimistic algorithm from Joulani et al. (2017) to the reformulation of Game over occupancy measures, we present the resulting algorithm, i.e., OMPO, in Alg. 3.

**Remark 5.** *In a partially observable Markov game, lifting the problem to the occupancy measures turns out to be fundamentally important for enabling each agent to learn a policy conditioned only on their own state. This is different from the standard literature on Markov Games (Daskalakis et al., 2020; Wei et al., 2021; Alacaoglu et al., 2022), which assumes that both agents share a common state.*

As the next theorem shows, in the ideal case where the updates can be computed exactly, Alg. 3 finds an $\epsilon$-approximate Nash equilibrium using fewer updates compared to Alg. 1 and to (Swamy et al., 2024, Algorithm 1). The proof can be found at Appx. D.4.

Table 1: Evaluation results on MT-bench-101 dataset. Mistral-7B-Instruct is selected as the base model. We can observe that both of the proposed algorithms `MPO` and `OMPO` considerably outperform the baseline in terms of the score (the higher the better).

| Model | Avg. | Perceptivity | | | | | Adaptability | | | | | | Interactivity | |
|---|---|---|---|---|---|---|---|---|---|---|---|---|---|---|
| | | Memory | Understanding | | Interference | | Rephrasing | | Reflection | | Reasoning | | Questioning | |
| | | CM | SI | AR | TS | CC | CR | FR | SC | SA | MR | GR | IC | PI |
| Base (Mistral-7B-Instruct) | 6.223 | 7.202 | 7.141 | 7.477 | 7.839 | 8.294 | 6.526 | 6.480 | 4.123 | 4.836 | 4.455 | 5.061 | 5.818 | 5.641 |
| DPO (iter=1) | 6.361 | 7.889 | 6.483 | 7.699 | 8.149 | 8.973 | 7.098 | 7.423 | 3.448 | **6.123** | 3.421 | 4.492 | 5.639 | 5.858 |
| DPO (iter=2) | 6.327 | 7.611 | 6.206 | 8.106 | 8.052 | 9.111 | 6.670 | 7.153 | 3.494 | 5.884 | 3.360 | 4.691 | 5.837 | 6.078 |
| DPO (iter=3) | 5.391 | 6.019 | 4.521 | 6.890 | 6.631 | 8.177 | 5.437 | 5.723 | 3.448 | 5.295 | 3.142 | 4.015 | 5.256 | 5.529 |
| SPPO (iter=1) | 6.475 | 7.432 | 7.464 | 7.714 | 8.353 | 8.580 | 6.917 | 6.714 | 4.136 | 5.055 | 4.403 | 5.400 | 6.036 | 5.966 |
| SPPO (iter=2) | 6.541 | 7.516 | 7.496 | 7.808 | 8.313 | 8.731 | 7.077 | 6.867 | 4.136 | 5.281 | 4.488 | 5.477 | 6.098 | 5.751 |
| SPPO (iter=3) | 6.577 | 7.575 | 7.547 | 7.944 | 8.365 | 8.797 | 7.040 | 6.865 | 4.442 | 5.185 | 4.346 | 5.394 | 6.092 | 5.906 |
| Step-DPO (iter=1) | 6.433 | 7.463 | 7.054 | 7.790 | 8.157 | 8.593 | 6.827 | 6.748 | 4.234 | 4.849 | 4.236 | 5.519 | 5.982 | 6.171 |
| Step-DPO (iter=2) | 6.553 | 7.616 | 7.043 | 7.925 | 8.147 | 8.662 | 6.790 | 6.878 | 4.331 | 5.048 | 4.366 | **5.734** | **6.391** | 6.254 |
| Step-DPO (iter=3) | 6.442 | 7.665 | 7.023 | 7.767 | 8.016 | 8.589 | 6.723 | 6.581 | 4.305 | 5.014 | 4.153 | 5.453 | 6.202 | **6.257** |
| MPO (iter=1) | 6.630 | 7.624 | **7.846** | 8.085 | 8.398 | 8.947 | 7.105 | 7.286 | 4.208 | 4.993 | 4.377 | 5.264 | 6.179 | 5.873 |
| MPO (iter=2) | 6.735 | 7.838 | 7.723 | 8.196 | **8.590** | 9.027 | 7.347 | 7.209 | 4.240 | 5.137 | 4.469 | 5.531 | 6.181 | 6.061 |
| MPO (iter=3) | 6.733 | **7.868** | 7.686 | **8.289** | 8.510 | 9.078 | 7.330 | 7.529 | **4.461** | 4.829 | 4.225 | 5.366 | 6.198 | 6.155 |
| OMPO(iter=2) | 6.736 | 7.733 | 7.723 | 8.257 | 8.478 | 9.122 | 7.300 | 7.421 | 4.123 | 5.288 | **4.506** | 5.513 | 6.179 | 5.923 |
| OMPO(iter=3) | **6.776** | 7.649 | 7.792 | 8.281 | 8.578 | **9.136** | **7.424** | **7.635** | 4.377 | 5.308 | 4.312 | 5.455 | 6.187 | 5.954 |

**Theorem 5** (Convergence of `OMPO`). *Consider Algorithm 3 and let us assume that the occupancy measure of the reference policy is uniformly lower bounded by $\underline{d}$. Moreover, let $\mathbb{D}$ be $1/\lambda$ strongly convex, i.e. $\mathbb{D}(p\|q) \geq \frac{\|p-q\|_1^2}{2\lambda}$. Then, by setting $T = \frac{10H \log \underline{d}^{-1}}{\beta\epsilon}$ and $\beta \leq \frac{1}{\sqrt{2\lambda}}$, we ensure that $(\pi^{\text{out}}, \pi^{\text{out}})$, i.e. the output of Algorithm 3 is an $\epsilon$-approximate Nash equilibrium. Therefore, we need at most $\frac{10H \log \underline{d}^{-1}}{\beta\epsilon}$ policy updates.*

In addition, not only Swamy et al. (2024, Algorithm 1) but also `OMPO` can be implemented using only one player since in a constant sum game, the max and min player produce the same iterates. The result is formalized as follows and the proof is deferred to Appx. D.5.

**Theorem 6.** *Consider a constant sum two-player Markov games with reward such that $r(s, a, s', a') = 1 - r(s', a', s, a)$, then for each $s_1 \in \text{supp}(\nu_1)$ the updates for $d$ in Alg. 3 coincides with the updates for the min player that uses the updates*

$$d_h^{t+1}(a|s) = \arg\min_{d \in \mathcal{F}_{s_1}} \beta \left\langle d, 2\mathbb{E}_{s',a' \sim d_h^t} r(s', a', \cdot, \cdot) - \mathbb{E}_{s',a' \sim d_h^{t-1}} r(s', a', \cdot, \cdot) \right\rangle + \mathbb{D}(d, d_h^t).$$

Furthermore, we can avoid the projection over the set $\mathcal{F}$ implementing this update on the policy space (see Appendix E). We achieve such results following the techniques developed in Bas-Serrano et al. (2021); Viano et al. (2022).

For the first iteration, we initialize $d_h^0$ to be equal to $d_h^1$ for all $h$. That is, at the first iteration, we use the same update rule as in `MPO`. After the first iteration, we apply similar techniques as in `MPO` by estimating the $Q$ function and we use a tunable parameter to approximate the $\log Z$ term. We illustrate the practical algorithm in Alg. 4.

## 5 EXPERIMENTS

In this section, we test the proposed algorithms with multi-turn conversations in MT-bench-101 (Bai et al., 2024). Additional experimental detail, ablation studies, and experiments on math reasoning tasks are deferred to Appx. F. We choose Mistral-7B-Instruct-v0.2 as the base model (Jiang et al., 2023). We use a pre-trained PairRM [4] as the preference oracle. Specifically, given two conversations $[s_h, a_h]$ and $[s_h', a_h']$, PairRM will return a score that indicates the probability that $[s_h, a_h]$

---

[4]https://huggingface.co/llm-blender/PairRM

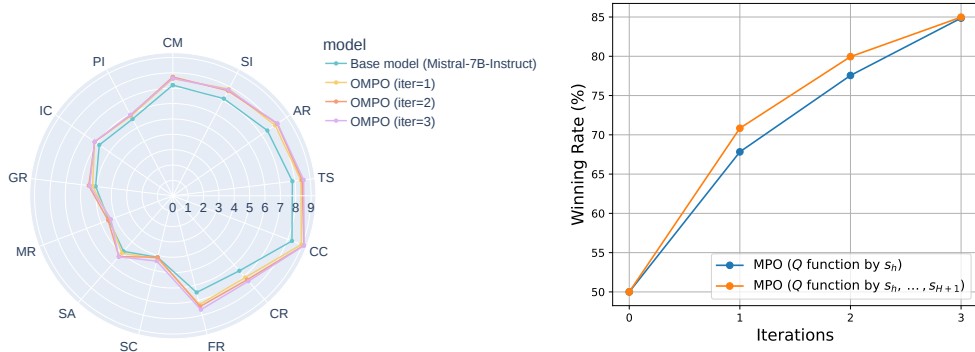

(a) Radar chart on different categories.  (b) Winning rate against the base model.

Figure 1: (a): Result of OMPO on the MT-bench-101 dataset; (b) Winning rate against the base model with different approximations for the $Q$ functions. When optimizing $a_h$ at the $h$ step, only considering the preference of $s_h$ is sufficient compared to using $s_h, \ldots, s_{H+1}$.

is better than $[s'_h, a'_h]$, which can be used to considered as the preference oracle $\mathbb{P}$ defined in the previous section. We select iterative DPO (Dong et al., 2024), iterative SPPO (Wu et al., 2024), and iterative Step-DPO as our baselines. For both iterative DPO and iterative SPPO, we sample $K = 5$ complete conversations starting from $s_1$, and estimate the winning rate $\mathbb{P}([s^k_{H+1}, a^k_{H+1}] \succ (s^{k'}_{H+1}, a^{k'}_{H+1})) \forall k, k' \in [K]$. Then we select both the best and worst conversations according to their winning rates against others, which is defined as $\frac{1}{K} \sum_{k'=1}^{K} \mathbb{P}([s^k_{H+1}, a^k_{H+1}] \succ [s^{k'}_{H+1}, a^{k'}_{H+1}])$ for the conversation $[s^k_{H+1}, a^k_{H+1}]$. Such a pair is used to train DPO while the winning rate is used to train SPPO. For both Step-DPO, MPO, and OMPO, we do the same strategy with starting at $s_h$. In MPO, and OMPO, we estimate $Q(s_h, a_h, s_h, a'_h)$ by $\mathbb{P}([s_h, a_h], [s_h, a'_h])$ to enhance the efficiency. For OMPO, the $Q^{\pi^t, \pi^{t-1}}$ term is estimated by calculating the winning rate between two answers (the best and the worst) generated by the current policy $\pi^t$ and the five answers previously generated by $\pi^{t-1}$, the $\tau$ is selected as zero. Each method is trained with epochs number selected from $\{1, 2\}$, learning rates from $\{5e\text{-}6, 5e\text{-}7\}$, and $\beta$ values from $\{0.1, 0.01, 0.001\}$. The final model is chosen based on the highest winning rate against the base model, as determined by the PairRM model. We use full-parameter fine-tuning for all methods with bf16 precision. A batch size of 64 is used. The maximum output length and maximum prompt length during training are both set as 2048. We use AdamW optimizer (Loshchilov & Hutter, 2019) and cosine learning rate schedule (Loshchilov & Hutter, 2017) with a warmup ratio of 0.1. Each round of dialogue is rated on a scale of 1 to 10 by GPT-4o mini, with the mean score reported for each dialogue. All methods are run for a total of 3 iterations. The results are summarized in Tab. 1, showing significant improvements over the baselines with the proposed MPO and OMPO approaches. In Fig. 1(a), we present the Radar chart on different categories and we can see that the proposed OMPO leads to improvements generally along the iterations. Fig. 1(b) shows that using the entire trajectory to estimate the $Q$ function can lead to subtle improvement at the first two iterations while it finally achieves a similar winning rate when compared to the one that only use one step.

## 6  CONCLUSION

This work presents a novel framework to enhance the preference alignment of large language models in multi-step settings by casting the alignment process as a two-player Markov game. We introduce novel algorithms based on natural actor-critic and optimistic online gradient descent, supported by both theoretical analysis and empirical results. However, the limitations of this work include the finite-horizon assumption in our theoretical framework, which may not fully capture real-world conversations or reasoning processes that often span with different steps instead of a fixed step $H$. Additionally, our practical algorithm requires querying a preference oracle, which may limit its applicability in cases where such preference oracles are unavailable or when collecting human feedback is costly. Future work should explore extending the theoretical framework to infinite-horizon settings and finding more scalable methods for gathering preference feedback.

## ETHICS STATEMENT

Our work focuses on algorithmic innovations related to reinforcement learning with human feedback. We do not create any new benchmarks for human preferences nor solicit human preferences for this study. As such, we do not expect any potential violations of ethical standards, including those concerning the use of human data. Our contributions are primarily methodological and theoretical analysis of the convergence, and we have taken care to ensure that our work complies with all relevant ethical guidelines.

## REPRODUCIBILITY STATEMENT

In this work, we have provided the details on the experimental setup and the description of the dataset at Sec. 5 and Appx. F.1. The dataset and language models used in this work are publicly available. The source code of MPO and OMPO will be made public in the camera-ready version. Regarding the theoretical results, we have clearly mentioned all of the assumptions, and all the complete proofs can be found at Appx. D and Appx. E.

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

CONTENTS OF THE APPENDIX

The Appendix is organized as follows:

## A    SYMBOLS AND NOTATION

We include the core symbols and notation in Tab. 2 to facilitate the understanding of our work.

Table 2: Core symbols and notations used in this paper.

| Symbol | Dimension(s) & range | Definition |
|---|---|---|
| $x_h$ | - | Prompt at step $h$ |
| $a_h$ | - | Answer (action) at step $h$ |
| $s_h$ | - | State at step $h$ |
| $s_1(s_h)$ | - | The only initial state that can lead to $s_h$ |
| $\pi$ | | Language model (policy) |
| $\nu_1$ | | Initial distribution of state $s_1$ |
| $d_h^\pi(s, a)$ | $[0, 1]$ | Occupancy measure of $\pi$ at stage $h$ |
| $f$ | | Transition function |
| $\Pr(s_h = s, a_h = a)$ | $[0, 1]$ | Joint probability of $s_h = a$ and $a_h = a$ |
| $o$ | $\{0, 1\}$ | Preference oracle |
| $\mathbb{P}([s, a], [s', a'])$ | $[0, 1]$ | Winning probability of $[s, a]$ against $[s', a']$ |
| $D(p \| q)$ | | KL divergence of two probability distributions $p$ and $q$ |
| $\mathbb{D}(p \| q)$ | | Bregman Divergences between two points $q$ and $p$. |
| $\mathcal{D}_t$ | | Dataset buffet at iteration t |
| $\Delta_\mathcal{X}$ | $[0, 1]^{|\mathcal{X}|}$ | Set of probability distributions over the set $\mathcal{X}$ |
| $\mathcal{O}, o, \Omega$ and $\Theta$ | - | Standard Bachmann–Landau order notation |

We additionally use a compact notation for representing the Bellman flow constraints. We denote by $E \in \mathbb{R}^{|\mathcal{S}| \times |\mathcal{A}||\mathcal{S}|}$ the matrix such that $(Ez)(s, a) = z(s)$ for all vectors $z \in \mathbb{R}^{|\mathcal{S}|}$. Additionally, we denote by $F$ the matrix such that $(Fz)(s, a) = \sum_{s'} f(s'|s, a)z(s')$ for all vectors $z \in \mathbb{R}^{|\mathcal{S}|}$.

## B    PRELIMINARY ON SINGLE-STEP RLHF

In this section, we review the earlier methods in single-step RLHF. Classical RLHF methods (Ziegler et al., 2019; Ouyang et al., 2022) assume that the preference oracle can be expressed by an underlying Bradley-Terry (BT) reward model (Bradley & Terry, 1952), i.e.,

$$\mathbb{P}([x_1, a_1] \succ [x_1, a'_1]) = \sigma(r(x_1, a_1) - r(x_1, a'_1)).$$

Thus, one can first learn a reward model and optimize the policy based on the following KL-constrained RL objective with PPO:

$$\pi^\star = \arg\max_\pi \mathbb{E}_{x_1 \sim \nu_1, a_1 \sim \pi(\cdot|x_1)}(r(x_1, a_1) - \beta D(\pi(\cdot|x_1) \| \pi_{\text{ref}}(\cdot|x_1))),$$

where $\beta$ is a parameter controlling the deviation from the reference model $\pi_{\text{ref}}$. Another line of work, e.g., DPO (Rafailov et al., 2023) avoids explicit reward modeling and optimizes the following objective over pair-wise preference data $(x_1, a_1^w, a_1^l)$.

$$\pi^\star = \arg\max_\pi \mathbb{E}_{(x_1, a_1^w, a_1^l) \sim \mathcal{D}} \left[ \log \sigma \left( \beta \log \frac{\pi(a_1^w|x_1)}{\pi_1(a_1^w|x_1)} - \beta \log \frac{\pi(a_1^l|x_1)}{\pi_1(a_1^l|x_1)} \right) \right].$$

More recently, several studies (Swamy et al., 2024; Munos et al., 2024; Wu et al., 2024; Zhang et al., 2024; Rosset et al., 2024) have circumvented the Bradley-Terry (BT) assumption by directly modeling the general oracle $\mathbb{P}$, avoiding the reliance on the reward model which is transitive. Specifically, the goal is to identify the Nash equilibrium (or von Neumann winner) of the following two-player constant-sum game:

$$(\pi^*, \pi^*) = \arg\max_{\pi} \min_{\pi'} \mathbb{E}_{x_1 \sim \nu_1, a_1 \sim \pi(\cdot|x_1), a_1' \sim \pi'(\cdot|x_1)} \mathbb{P}([x_1, a_1] \succ [x_1, a_1']).$$

## C ADDITIONAL DISCUSSION ON RELATED WORK

### C.1 RELATED WORK ON TOKEN-LEVEL PREFERENCE OPTIMIZATION

A line of work formulates the alignment of contextual bandit problems in LLMs (Example.1) from token-level MDPs perspective (Rafailov et al., 2024; Zeng et al., 2024; Liu et al., 2024a). In Rafailov et al. (2024), by defining the reward at each token before the terminal token as the generation likelihood and using the maximum entropy RL objective, the authors derive the original objective of DPO from a new perspective that incorporates token-level rewards. Zeng et al. (2024) assume that the reward for a response can be decomposed into token-level rewards at each token. Then they design a token-level objective function based on Trust Region Policy Optimization, adding token-level KL divergence constraints to the DPO objective in the final algorithm. More recently, Liu et al. (2024a) study how the difference in average rewards between chosen and rejected responses affects the optimization stability, designing a new algorithm where importance sampling weights are assigned to each token-level reward. There are two main differences between the multi-step alignment approach in our work and those in previous work. First, while Rafailov et al. (2024); Zeng et al. (2024); Liu et al. (2024a) develop alignment methods based on the Bradley-Terry model with transitive rewards, our framework is motivated by a two-player game with relative rewards. Secondly, although Rafailov et al. (2024); Zeng et al. (2024); Liu et al. (2024a) formulate the alignment process as an MDP, their final objective is tailored to a contextual bandit problem in LLMs. In contrast, our objective is designed for a multi-step alignment problem, suited for multi-turn conversation or chain-of-thought reasoning.

### C.2 DISCUSSION ON THE DIFFERENCE FROM SPPO

Next, we elaborate on the difference with SPPO (Wu et al., 2024) below: Firstly, the theoretical analysis of the proposed MPO differs from that of SPPO due to differences in the settings. SPPO considers the contextual bandit problem and builds its analysis based on the game matrix from Freund & Schapire (1999). In our case, however, we frame the problem as a Markov game and employ a distinct theoretical analysis apart from Freund & Schapire (1999). Specifically, in our proof, we (i) use the performance difference lemma to rewrite the global regret as weighted average of local regrets and (ii) control the local regrets with multiplicative weights updates. Secondly, a new algorithm, OMPO, is developed in this work with a novel theoretical guarantee. In the case where the horizon $H = 1$, the update of OMPO reduces to

$$\pi^{t+1}(a|s) \propto \pi^t(a|s) \exp\left[\beta(2\mathbb{P}(a \succ \pi^t(\cdot|s)) - \mathbb{P}(a \succ \pi^{t-1}(\cdot|s)))\right],$$

while the update of SPPO is

$$\pi^{t+1}(a|s) \propto \pi^t(a|s) \exp\left[\beta(\mathbb{P}(a \succ \pi^t(\cdot|s)))\right].$$

As a result, OMPO enables $\mathcal{O}(\epsilon^{-1})$ policy updates to converge to an $\epsilon$-approximate Nash equilibrium instead of $\mathcal{O}(\epsilon^{-2})$, according to our theoretical analysis.

## D PROOFS

### D.1 PROOF OF LEMMA 1

*Proof.* By the definition of the state action value function for the policy pair $(\pi, \pi')$ we have that

$$Q_h^{\pi, \pi'}(s, a, s', a') = r(s, a, s', a') + \mathbb{E}\left[\sum_{h'=h+1}^{H} r(s_{h'}, a_{h'}, s_{h'}', a_{h'}')\right].$$

Now, using tower property of the expectation we have that

$$Q_h^{\pi,\pi'}(s,a,s',a')$$

$$= r(s,a,s',a') + \mathbb{E}_{s''\sim f(\cdot|s,a),\bar{s}\sim f(\cdot|s',a')}\Big[\mathbb{E}\Big[\sum_{h'=h+1}^{H} r(s_{h'},a_{h'},s'_{h'},a'_{h'})|s_{h+1}=s'',s'_{h+1}=\bar{s}\Big]\Big]$$

$$= r(s,a,s',a') + \mathbb{E}_{s''\sim f(\cdot|s,a),\bar{s}\sim f(\cdot|s',a')}\Big[V^{\pi,\pi'}(s'',\bar{s})\Big],$$

where the last equality follows from the definition of the state value function. $\qquad\square$

## D.2 Proof of Lemma 2

*Proof.* Let us consider the Bellman equation in vectorial form for the policy pair $(\pi',\bar{\pi})$, that is

$$r_h + FV_{h+1}^{\pi',\bar{\pi}} = Q_h^{\pi',\bar{\pi}},$$

where $F$ denoted the transition matrix induced by the transition function $f : \mathcal{S}^2 \times \mathcal{A} \to \Delta_{\mathcal{S}\times\mathcal{S}}$. Now, multiplying by the occupancy measure of the policy pair $(\pi,\bar{\pi})$ at stage $h$ we obtain

$$\langle d_h^{\pi,\bar{\pi}}, r_h\rangle + \Big\langle d_h^{\pi,\bar{\pi}}, FV_{h+1}^{\pi',\bar{\pi}}\Big\rangle = \Big\langle d_h^{\pi,\bar{\pi}}, Q_h^{\pi',\bar{\pi}}\Big\rangle.$$

At this point, using the Bellman flow constraints Puterman (1994), it holds that

$$F^T d_h^{\pi,\bar{\pi}} = E^T d_{h+1}^{\pi,\bar{\pi}},$$

where $E \in \mathbb{R}^{|\mathcal{S}|^2|\mathcal{A}|\times|\mathcal{S}|^2}$ such that $(E^T V)(s,a) = V(s)$ for all $V \in \mathbb{R}^{|\mathcal{S}|^2}$. Plugging this equality in the Bellman equation above we obtain

$$\langle d_h^{\pi,\bar{\pi}}, r_h\rangle + \Big\langle d_{h+1}^{\pi,\bar{\pi}}, EV_{h+1}^{\pi',\bar{\pi}}\Big\rangle = \Big\langle d_h^{\pi,\bar{\pi}}, Q_h^{\pi',\bar{\pi}}\Big\rangle.$$

Now, subtracting on both sides $\Big\langle d_h^{\pi,\bar{\pi}}, EV_h^{\pi',\bar{\pi}}\Big\rangle$ and rearranging, it holds that

$$\langle d_h^{\pi,\bar{\pi}}, r_h\rangle + \Big\langle d_{h+1}^{\pi,\bar{\pi}}, EV_{h+1}^{\pi',\bar{\pi}}\Big\rangle - \Big\langle d_h^{\pi,\bar{\pi}}, EV_h^{\pi',\bar{\pi}}\Big\rangle = \Big\langle d_h^{\pi,\bar{\pi}}, Q_h^{\pi',\bar{\pi}} - EV_h^{\pi',\bar{\pi}}\Big\rangle.$$

After this, taking sum from $h=1$ to $H$ and recognizing that for all policy pairs $(\pi,\pi')$ it holds that $V_{H+1}^{\pi,\pi'}=0$, it holds that

$$\sum_{h=1}^{H}\langle d_h^{\pi,\bar{\pi}}, r_h\rangle - \Big\langle d_1^{\pi,\bar{\pi}}, EV_1^{\pi',\bar{\pi}}\Big\rangle = \sum_{h=1}^{H}\Big\langle d_h^{\pi,\bar{\pi}}, Q_h^{\pi',\bar{\pi}} - EV_h^{\pi',\bar{\pi}}\Big\rangle.$$

Then, notice that for all policies $\pi,\bar{\pi}$ it holds that $\sum_{h=1}^{H}\langle d_h^{\pi,\bar{\pi}}, r_h\rangle = \langle\nu_1, V^{\pi,\bar{\pi}}\rangle$. Plugging in these observations, we get

$$\Big\langle\nu_1, V^{\pi,\bar{\pi}} - V^{\pi',\bar{\pi}}\Big\rangle = \sum_{h=1}^{H}\Big\langle d_h^{\pi,\bar{\pi}}, Q_h^{\pi',\bar{\pi}} - EV_h^{\pi',\bar{\pi}}\Big\rangle.$$

Therefore, expanding the expectation, and noticing that $d_h^{\pi,\bar{\pi}}(s,a,s',a'|s_1) = d_h^{\pi}(s,a|s_1)d_h^{\bar{\pi}}(s',a'|s_1)$ for all $h,s,a,s',a'$ and conditioning $s_1$, we get that

$$\Big\langle\nu_1, V^{\pi,\bar{\pi}} - V^{\pi',\bar{\pi}}\Big\rangle$$

$$= \mathbb{E}_{s_1\sim\nu_1}\sum_{h=1}^{H}\mathbb{E}_{s\sim d_h^{\pi}|s_1}\Big[\Big\langle\mathbb{E}_{s',a'\sim d_h^{\bar{\pi}}|s_1}Q_h^{\pi',\bar{\pi}}(s,\cdot,s',a'), \pi_h(\cdot|s,s_1) - \pi'_h(\cdot|s,s_1)\Big\rangle\Big].$$

$\qquad\square$

### D.3 PROOF OF THM. 4

*Proof.* We set $\bar{\pi}_h^T(a_h|s_h) = \frac{\sum_{t=1}^T d_h^{\pi^t}(s_h,a_h)}{\sum_{t=1}^T d_h^{\pi^t}(s_h)}$, where $d(s)$ is the marginal distribution of $d(s,a)$ on state $s$, and $\bar{\pi}^T = (\bar{\pi}_h^T)_{h=1}^H$. We shows that $d_h^{\bar{\pi}^T} = \frac{1}{T}\sum_{t=1}^T d_h^{\pi^t}$ by induction. $h = 1$ holds by definition. Assuming on step $h$, the equation holds, we have

$$
\begin{aligned}
d_{h+1}^{\bar{\pi}^T}(s_{h+1}, a_{h+1}) &= d_{h+1}^{\bar{\pi}^T}(s_{h+1})\bar{\pi}_{h+1}^T(a_{h+1}|s_{h+1}) \\
&= \sum_{s_h, a_h \sim \bar{\pi}_h^T(\cdot|s_h)} d_h^{\bar{\pi}^T}(s_h, a_h) f(s_{h+1}|s_h, a_h)\bar{\pi}_{h+1}^T(a_{h+1}|s_{h+1}) \\
&= \sum_{s_h, a_h \sim \bar{\pi}_h^T(\cdot|s_h)} \frac{1}{T}\sum_{t=1}^T d_h^{\pi^t}(s_h, a_h) f(s_{h+1}|s_h, a_h)\bar{\pi}_{h+1}^T(a_{h+1}|s_{h+1}) \\
&= \frac{1}{T}\sum_{t=1}^T d_{h+1}^{\pi^t}(s_{h+1})\bar{\pi}_{h+1}^T(a_{h+1}|s_{h+1}) \\
&= \frac{1}{T}\sum_{t=1}^T d_{h+1}^{\pi^t}(s_{h+1}, a_{h+1}),
\end{aligned}
$$

where the last equation holds by definition of $\bar{\pi}_{h+1}^T$. Therefore, $h+1$ holds, and the $\bar{\pi}^T$ satisfy all equations for $h \in [H]$.

Using the value difference Lemma 2 we have that for any $\pi^\star \in \Pi$

$$
\begin{aligned}
&\left\langle \nu_1, V^{\pi^\star, \pi^t} - V^{\pi^t, \pi^t} \right\rangle \\
&= \mathbb{E}_{s_1 \sim \nu_1} \sum_{h=1}^H \mathbb{E}_{s \sim d_h^{\pi^\star}|s_1}\left[ \left\langle \mathbb{E}_{s', a' \sim d_h^{\pi^t}|s_1} Q_h^{\pi^t, \pi^t}(s, \cdot, s', a'), \pi_h^\star(\cdot|s) - \pi_h^t(\cdot|s) \right\rangle \right].
\end{aligned}
$$

Therefore, summing over $t$ from $t = 1$ to $T$ we obtain

$$
\begin{aligned}
&\sum_{t=1}^T \left\langle \nu_1, V^{\pi^\star, \pi^t} - V^{\pi^t, \pi^t} \right\rangle \\
&= \mathbb{E}_{s_1 \sim \nu_1} \sum_{h=1}^H \mathbb{E}_{s \sim d_h^{\pi^\star}|s_1}\left[ \sum_{t=1}^T \left\langle \mathbb{E}_{s', a' \sim d_h^{\pi^t}|s_1} Q_h^{\pi^t, \pi^t}(s, \cdot, s', a'), \pi_h^\star(\cdot|s) - \pi_h^t(\cdot|s) \right\rangle \right].
\end{aligned}
$$

Therefore, we need to control the local regrets at each state $s$ with loss $\ell_h^t(s, s_1) := \mathbb{E}_{s', a' \sim d_h^{\pi^t}|s_1} Q_h^{\pi^t, \pi^t}(s, \cdot, s', a')$. To this end, we can invoke a standard convergence result for online mirror descent (Orabona, 2023, Theorem 6.10) we obtain that at each state we have

$$
\sum_{t=1}^T \left\langle \ell_h^t(s, s_1), \pi^\star(\cdot|s) - \pi^t(\cdot|s) \right\rangle \le \frac{D(\pi^\star(\cdot|s), \pi^1(\cdot|s))}{\beta} + \beta\sum_{t=1}^T \|\ell_h^t(s, s_1)\|_\infty^2.
$$

Now, noticing that we have $\|\ell_h^t(s, s_1)\|_\infty \le H$ it holds that

$$
\sum_{t=1}^T \left\langle \ell_h^t(s), \pi_h^\star(\cdot|s) - \pi_h^t(\cdot|s) \right\rangle \le \frac{D(\pi_h^\star(\cdot|s), \pi_h^1(\cdot|s))}{\beta} + \beta T H^2.
$$

Finally, using the assumption that $\pi^1(a|s) \ge \underline{\pi}$ for all $s, a \in \mathcal{S} \times \mathcal{A}$ it holds that $D(\pi^\star(\cdot|s), \pi^1(\cdot|s)) \le \log \underline{\pi}^{-1}$. Therefore, choosing $\beta = \sqrt{\frac{\log \underline{\pi}^{-1}}{TH^2}}$ it holds that

$$
\sum_{t=1}^T \left\langle \ell_h^t(s, s_1), \pi^\star(\cdot|s) - \pi^t(\cdot|s) \right\rangle \le 2H\sqrt{T \log \underline{\pi}^{-1}}.
$$

Thus, we conclude that

$$\sum_{t=1}^{T} \left\langle \nu_1, V^{\pi^\star,\pi^t} - V^{\pi^t,\pi^t} \right\rangle \le 2H^2 \sqrt{T \log \underline{\pi}^{-1}}.$$

By the antisimmetry of the game, the same proof steps

$$\sum_{t=1}^{T} \left\langle \nu_1, V^{\pi^t,\pi^t} - V^{\pi^t,\bar{\pi}^\star} \right\rangle \le 2H^2 \sqrt{T \log \underline{\pi}^{-1}}.$$

Therefore, it holds that for all $\pi^\star, \bar{\pi}^\star \in \Pi$

$$\sum_{t=1}^{T} \left\langle \nu_1, V^{\pi^\star,\pi^t} - V^{\pi^t,\pi^\star} \right\rangle \le 4H^2 \sqrt{T \log \underline{\pi}^{-1}}.$$

Then, define $\bar{\pi}^T$ the trajectory level mixture policy as in Swamy et al. (2024), i.e. such that $d_h^{\bar{\pi}^T} = \frac{1}{T} \sum_{t=1}^{T} d_h^{\pi^t}$ for all stages $h \in [H]$. This implies that $V^{\bar{\pi}^T,\pi^\star} = \frac{1}{T} \sum_{t=1}^{T} V^{\pi^t,\pi^\star}$, and $V^{\pi^\star,\bar{\pi}^T} = \frac{1}{T} \sum_{t=1}^{T} V^{\pi^\star,\pi_t}$.

Therefore, we have that

$$\left\langle \nu_1, V^{\pi^\star,\bar{\pi}^T} - V^{\bar{\pi}^T,\bar{\pi}^\star} \right\rangle \le 4H^2 \sqrt{\frac{\log \underline{\pi}^{-1}}{T}}.$$

Finally, selecting $\pi^\star = \left\langle \nu_1, \arg\max_{\pi \in \Pi} V^{\pi,\bar{\pi}^T} \right\rangle$ and $\bar{\pi}^\star = \left\langle \nu_1, \arg\min_{\pi \in \Pi} V^{\bar{\pi}^T,\pi} \right\rangle$, we obtain that

$$\max_{\pi \in \Pi} \left\langle \nu_1, V^{\pi,\bar{\pi}^T} \right\rangle - \min_{\pi \in \Pi} \left\langle \nu_1, V^{\bar{\pi}^T,\pi} \right\rangle \le 4H^2 \sqrt{\frac{\log \underline{\pi}^{-1}}{T}}.$$

This implies that

$$\left\langle \nu_1, V^{\bar{\pi}^T,\bar{\pi}^T} \right\rangle - \min_{\pi \in \Pi} \left\langle \nu_1, V^{\bar{\pi}^T,\pi} \right\rangle \le 4H^2 \sqrt{\frac{\log \underline{\pi}^{-1}}{T}},$$

and

$$\max_{\pi \in \Pi} \left\langle \nu_1, V^{\pi,\bar{\pi}^T} \right\rangle - \left\langle \nu_1, V^{\bar{\pi}^T,\bar{\pi}^T} \right\rangle \le 4H^2 \sqrt{\frac{\log \underline{\pi}^{-1}}{T}},$$

Therefore, setting $T = \frac{16H^4 \log \underline{\pi}^{-1}}{\epsilon^2}$ we obtain an $\epsilon$-approximate Nash equilibrium. $\qquad\square$

### D.4 PROOF OF THEOREM 5

*Proof.* The optimization problem

$$\arg\max_{d \in \tilde{\mathcal{F}}} \min_{d' \in \tilde{\mathcal{F}}} \mathbb{E}_{s_1 \sim \nu_1} \sum_{h=1}^{H} \sum_{s,a,s',a'} d_h(s,a|s_1) r(s,a,s',a') d'_h(s',a'|s_1)$$

can be carried out individually over possible initial states. That is for each $s_1 \in \mathrm{supp}(\nu_1)$ we aim at solving

$$\arg\max_{d \in \mathcal{F}_{s_1}} \min_{d' \in \mathcal{F}_{s_1}} \sum_{h=1}^{H} \sum_{s,a,s',a'} d_h(s,a|s_1) r(s,a,s',a') d'_h(s',a'|s_1)$$

To this end for any $s_1$, we consider $\phi_h^t \in \mathcal{F}$ and $\psi_h^t \in \mathcal{F}$ which are generated by the following updates

$$\phi_h^{t+1} = \arg\max_{\phi \in \mathcal{F}_{s_1}} \beta \left\langle \phi, 2\mathbb{E}_{s',a' \sim \psi^t} r_h(\cdot,\cdot,s',a') - \mathbb{E}_{s',a' \sim \psi^{t-1}} r_h(\cdot,\cdot,s',a') \right\rangle - \mathbb{D}(\phi, \phi_h^t),$$

and

$$\psi_h^{t+1} = \arg\min_{\psi \in \mathcal{F}_{s_1}} \beta \left\langle \psi, 2\mathbb{E}_{s',a' \sim \phi^t} r_h(s',a',\cdot,\cdot) - \mathbb{E}_{s',a' \sim \phi^{t-1}} r_h(s',a',\cdot,\cdot) \right\rangle + \mathbb{D}(\psi, \psi_h^t),$$

In order to prove convergence to an $\epsilon$-approximate Nash equilibrium, we need to control the quantity

$$\text{Gap}_{s_1} = \frac{1}{T}\sum_{h=1}^{H}\sum_{t=1}^{T}\left\langle\theta_h^t, \phi_h^t - \phi_h^\star\right\rangle + \frac{1}{T}\sum_{h=1}^{H}\sum_{t=1}^{T}\left\langle\zeta_h^t, \psi_h^t - \psi_h^\star\right\rangle,$$

for $\theta_h^t(s,a) = \sum_{s',a'}\psi_h^t(s',a')r_h(s,a,s',a')$ and $\zeta_h^t(s',a') = -\sum_{s,a}\phi_h^t(s,a)r_h(s,a,s',a')$. At this point, we bound the local regret term with the OMPO update. We have that for any $\phi_h \in \mathcal{F}$

$$\begin{aligned}
\beta\left\langle 2\theta_h^t - \theta_h^{t-1}, \phi_h - \phi_h^{t+1}\right\rangle &= \beta\left\langle \theta_h^t - \theta_h^{t+1}, \phi_h - \phi_h^{t+1}\right\rangle \\
&\quad + \beta\left\langle \theta_h^t + \theta_h^{t+1} - \theta_h^{t-1}, \phi_h - \phi_h^{t+1}\right\rangle \\
&= \beta\left\langle \theta_h^t - \theta_h^{t+1}, \phi_h - \phi_h^{t+1}\right\rangle \\
&\quad + \beta\left\langle \theta_h^t - \theta_h^{t-1}, \phi_h - \phi_h^t\right\rangle \\
&\quad + \beta\left\langle \theta_h^t - \theta_h^{t-1}, \phi_h^t - \phi_h^{t+1}\right\rangle \\
&\quad + \beta\left\langle \theta_h^{t+1}, \phi_h - \phi_h^{t+1}\right\rangle.
\end{aligned}$$

At this point, we work on the third summand above

$$\beta\left\langle \theta_h^t - \theta_h^{t-1}, \phi_h^t - \phi_h^{t+1}\right\rangle \leq \beta^2\lambda\|\theta_h^t - \theta_h^{t-1}\|_\infty^2 + \frac{1}{4\lambda}\|\phi_h^t - \phi_h^{t+1}\|_1^2.$$

In addition, we have that $\|\theta_h^t - \theta_h^{t-1}\|_\infty \leq \|\psi_h^t - \psi_h^{t-1}\|_1$ and we can apply the $1/\lambda$ strong convexity of $\mathbb{D}$, we obtain

$$\beta\left\langle \theta_h^t - \theta_h^{t-1}, \phi_h^t - \phi_h^{t+1}\right\rangle \leq \lambda\beta^2\|\psi_h^t - \psi_h^{t-1}\|_1^2 + \frac{1}{2}\mathbb{D}(\phi_h^{t+1}, \phi_h^t).$$

On the other hand, by the three point identity we have that for all $\phi \in \mathcal{F}$

$$\mathbb{D}(\phi_h, \phi_h^{t+1}) = \mathbb{D}(\phi_h, \phi_h^t) - \mathbb{D}(\phi_h^{t+1}, \phi_h^t) + \left\langle\nabla\mathbb{D}(\phi_h^{t+1}, \phi_h^t), \phi_h^{t+1} - \phi_h\right\rangle$$

Then, using the property of the update rule, we obtain that

$$\left\langle\nabla\mathbb{D}(\phi_h^{t+1}, \phi_h^t), \phi_h^{t+1} - \phi_h\right\rangle \leq \beta\left\langle 2\theta_h^t - \theta_h^{t-1}, \phi_h - \phi_h^{t+1}\right\rangle.$$

Putting all the pieces together we have that

$$\begin{aligned}
\mathbb{D}(\phi_h, \phi_h^{t+1}) &\leq \mathbb{D}(\phi_h, \phi_h^t) - \mathbb{D}(\phi_h^{t+1}, \phi_h^t) + \beta\left\langle 2\theta_h^t - \theta_h^{t-1}, \phi_h - \phi_h^{t+1}(\cdot|s)\right\rangle \\
&\leq \mathbb{D}(\phi_h, \phi_h^t) - \mathbb{D}(\phi_h^{t+1}, \phi_h^t) \\
&\quad + \beta\left\langle \theta_h^t - \theta_h^{t+1}, \phi_h - \phi_h^{t+1}\right\rangle \\
&\quad + \beta\left\langle \theta_h^t - \theta_h^{t-1}, \phi_h - \phi_h^t\right\rangle \\
&\quad + \beta^2\|\psi_h^t - \psi_h^{t-1}\|_1^2 + \frac{1}{2}\mathbb{D}(\phi_h^{t+1}, \phi_h^t) \\
&\quad + \beta\left\langle \theta_h^{t+1}, \phi_h - \phi_h^{t+1}\right\rangle.
\end{aligned}$$

Now, rearranging the terms we get

$$\begin{aligned}
\beta\left\langle \theta_h^{t+1}, \phi_h - \phi_h^{t+1}\right\rangle &\leq \mathbb{D}(\phi_h, \phi_h^t) - \mathbb{D}(\phi_h, \phi_h^{t+1}) - \frac{1}{2}\mathbb{D}(\phi_h^{t+1}, \phi_h^t) \\
&\quad + \beta\left\langle \theta_h^t - \theta_h^{t+1}, \phi_h - \phi_h^{t+1}\right\rangle \\
&\quad + \beta\left\langle \theta_h^t - \theta_h^{t-1}, \phi_h - \phi_h^t\right\rangle \\
&\quad + \beta^2\lambda\|\psi_h^t - \psi_h^{t-1}\|_1^2.
\end{aligned}$$

Now, denoting $\Phi_\phi^t := \mathbb{D}(\phi_h, \phi_h^t) + \beta\left\langle\theta_h^t - \theta_h^{t-1}, \phi_h - \phi_h^t\right\rangle$ and summing over $t$ we obtain

$$\beta\sum_{t=1}^{T}\left\langle\theta_h^t, \phi_h - \phi_h^t\right\rangle \leq \sum_{t=1}^{T}\Phi_\phi^{t-1} - \Phi_\phi^t - \frac{1}{2}\sum_{t=1}^{T}\mathbb{D}(\phi_h^t, \phi_h^{t-1}) + \beta^2\lambda\sum_{t=1}^{T}\|\psi_h^{t-1} - \psi_h^{t-2}\|_1^2.$$

Similarly we get

$$\beta \sum_{t=1}^{T} \langle \zeta^t(s,\cdot), \psi_h^t - \psi_h^t \rangle \leq \sum_{t=1}^{T} \Phi_\psi^{t-1} - \Phi_\psi^t - \frac{1}{2} \sum_{t=1}^{T} \mathbb{D}(\psi_h^t, \psi_h^{t-1}) + \beta^2 \lambda \sum_{t=1}^{T} \|\phi_h^{t-1} - \psi_h^{t-2}\|_1^2.$$

Now, using $1/\lambda$ strong convexity of $\mathbb{D}$ and summing the two terms we have that

$$\beta T \mathrm{Gap}_{s_1, h} \leq \Phi^0 - \Phi^{T-1} - \frac{1}{2} \sum_{t=1}^{T} (\mathbb{D}(\psi_h^t, \psi_h^{t-1}) + \mathbb{D}(\phi_h^t, \phi_h^{t-1}))$$

$$+ 2\beta^2 \lambda \sum_{t=1}^{T} (\mathbb{D}(\psi_h^{t-1}, \psi_h^{t-2}) + \mathbb{D}(\phi_h^{t-1}, \phi_h^{t-2})),$$

with $\Phi^t = \Phi_\phi^t + \Phi_\psi^t$. At this point, setting $\beta \leq \frac{1}{\sqrt{2\lambda}}$, we obtain a telescopic sum

$$\beta T \mathrm{Gap}_{s_1, h}$$

$$\leq \Phi^0 - \Phi^{T-1} - \frac{1}{2} \sum_{t=1}^{T} (\mathbb{D}(\psi_h^t, \psi_h^{t-1}) + \mathbb{D}(\phi_h^t, \phi_h^{t-1}) - \mathbb{D}(\psi_h^{t-1}, \psi_h^{t-2}) - \mathbb{D}(\phi_h^{t-1}, \phi_h^{t-2}))$$

$$\leq \Phi^0 - \Phi^{T-1} + \frac{1}{2} \left( \mathbb{D}(\psi_h^1, \psi_h^0) + \mathbb{D}(\phi_h^1, \phi_h^0) \right).$$

Now recalling that by assumption the occupancy measure of the reference policy is lower bounded, i.e. $d^{\pi^1} \geq \underline{d}$, we can upper bound $\Phi^0 - \Phi^T \leq 2 \log \underline{d}^{-1} + 8\beta$ that allows to conclude that for all $n \in [N]$ and setting $\psi_h^0 = \psi_h^1$ and $\phi_h^1 = \phi_h^0$,

$$\mathrm{Gap}_{s_1, h} \leq \frac{2 \log \underline{d}^{-1} + 8\beta}{\beta T} \leq \frac{10 \log \underline{d}^{-1}}{\beta T}.$$

Now, notice that Gap can be rewritten as

$$\mathrm{Gap}_{s_1} = \sum_{h=1}^{H} \mathrm{Gap}_{s_1, h}$$

$$= \frac{1}{T} \sum_{t=1}^{T} \sum_{h=1}^{H} \sum_{s,a,s',a'} \psi_h^\star(s', a') r_h(s, a, s', a') \phi_h^t(s, a)$$

$$- \frac{1}{T} \sum_{t=1}^{T} \sum_{h=1}^{H} \sum_{s,a,s',a'} \psi_h^t(s', a') r_h(s, a, s', a') \phi_h^\star(s, a)$$

$$= \sum_{h=1}^{H} \sum_{s,a,s',a'} \psi_h^\star(s', a') r_h(s, a, s', a') \frac{1}{T} \sum_{t=1}^{T} \phi_h^t(s, a)$$

$$- \sum_{h=1}^{H} \sum_{s,a,s',a'} \frac{1}{T} \sum_{t=1}^{T} \psi_h^t(s', a') r_h(s, a, s', a') \phi_h^\star(s, a)$$

$$= \sum_{h=1}^{H} \sum_{s,a,s',a'} \psi_h^\star(s', a') r_h(s, a, s', a') \bar{\phi}_h(s, a) - \sum_{h=1}^{H} \sum_{s,a,s',a'} \bar{\psi}_h(s', a') r_h(s, a, s', a') \phi_h^\star(s, a).$$

At this point, let us define $\pi_\phi^{\mathrm{out}}(a|s) = \frac{\bar{\phi}(s,a)}{\sum_a \bar{\phi}(s,a)}$ and $\pi_\psi^{\mathrm{out}}(a|s) = \frac{\bar{\psi}(s,a)}{\sum_a \bar{\psi}(s,a)}$. For such policies and by appropriate choice for $\psi^\star$ and $\phi^\star$ it follows that

$$\mathrm{Gap}_{s_1} = \max_\psi V^{\pi_\phi^{\mathrm{out}}, \psi}(s_1) - \min_\phi V^{\phi, \pi_\psi^{\mathrm{out}}}(s_1).$$

By the bound on $\mathrm{Gap}_{s_1}$ for each $s_1 \in \mathrm{supp}(\nu_1)$, it follows that

$$\left\langle \nu_1, \max_\psi V^{\pi_\phi^{\mathrm{out}}, \psi} - \min_\phi V^{\phi, \pi_\psi^{\mathrm{out}}} \right\rangle = \mathbb{E}_{s_1 \sim \nu_1} \mathrm{Gap}_{s_1} \leq \frac{10 H \log \underline{d}^{-1}}{\beta T},$$

therefore $T \geq \frac{10 H \log \underline{d}^{-1}}{\beta \epsilon}$. The proof is concluded invoking Thm. 6 that ensures that the policies $\pi_\psi^{\mathrm{out}}$ and $\pi_\phi^{\mathrm{out}}$ coincide. $\qquad \square$

### D.5 PROOF OF THEOREM 6

*Proof.* Let us consider two players performing the following updates

$$\phi_h^{t+1} = \arg\max_{\phi \in \mathcal{F}_{s_1}} \beta \left\langle \phi, 2\mathbb{E}_{s',a' \sim \psi^t} r_h(\cdot, \cdot, s', a') - \mathbb{E}_{s',a' \sim \psi^{t-1}} r_h(\cdot, \cdot, s', a') \right\rangle - \mathbb{D}(\phi, \phi_h^t),$$

and

$$\psi_h^{t+1} = \arg\min_{\psi \in \mathcal{F}_{s_1}} \beta \left\langle \psi, 2\mathbb{E}_{s',a' \sim \phi^t} r_h(s', a', \cdot, \cdot) - \mathbb{E}_{s',a' \sim \phi^{t-1}} r_h(s', a', \cdot, \cdot) \right\rangle + \mathbb{D}(\psi, \psi_h^t).$$

The goal is to proof that the iterates generated by the two updates are identical. We will prove this fact by induction. The base case holds by initialization which gives $\phi_h^0 = \psi_h^0$ for all $h \in [H]$. Then, let us assume by the induction step that $\psi_h^t = \phi_h^t$ for all $h \in [H]$, then

$$\phi_h^{t+1}$$

$$= \arg\max_{\phi \in \mathcal{F}_{s_1}} \beta \left\langle \phi, 2\mathbb{E}_{s',a' \sim \psi^t} r_h(\cdot, \cdot, s', a') - \mathbb{E}_{s',a' \sim \psi^{t-1}} r_h(\cdot, \cdot, s', a') \right\rangle - \mathbb{D}(\phi, \phi_h^t)$$

$$= \arg\max_{\phi \in \mathcal{F}_{s_1}} \beta \left\langle \phi, -2\mathbb{E}_{s',a' \sim \psi^t} r_h(s', a', \cdot, \cdot) + \mathbb{E}_{s',a' \sim \psi^{t-1}} r_h(s', a', \cdot, \cdot) \right\rangle - \mathbb{D}(\phi, \phi_h^t) + \beta \left\langle \phi, \mathbf{1} \right\rangle$$

(Antisymmetric Reward)

$$= \arg\max_{\phi \in \mathcal{F}_{s_1}} \beta \left\langle \phi, -2\mathbb{E}_{s',a' \sim \psi^t} r_h(s', a', \cdot, \cdot) + \mathbb{E}_{s',a' \sim \psi^{t-1}} r_h(s', a', \cdot, \cdot) \right\rangle - \mathbb{D}(\phi, \phi_h^t) + \beta$$

(Normalization of $\phi$)

$$= \arg\max_{\phi \in \mathcal{F}_{s_1}} \beta \left\langle \phi, -2\mathbb{E}_{s',a' \sim \psi^t} r_h(s', a', \cdot, \cdot) + \mathbb{E}_{s',a' \sim \psi^{t-1}} r_h(s', a', \cdot, \cdot) \right\rangle - \mathbb{D}(\phi, \phi_h^t)$$

($\beta$ does not depend on $\phi$)

$$= \arg\max_{\phi \in \mathcal{F}_{s_1}} \beta \left\langle \phi, -2\mathbb{E}_{s',a' \sim \phi^t} r_h(s', a', \cdot, \cdot) + \mathbb{E}_{s',a' \sim \phi^{t-1}} r_h(s', a', \cdot, \cdot) \right\rangle - \mathbb{D}(\phi, \psi_h^t)$$

(Inductive Hypothesis)

$$= \arg\min_{\psi \in \mathcal{F}_{s_1}} \beta \left\langle \psi, 2\mathbb{E}_{s',a' \sim \phi^t} r_h(s', a', \cdot, \cdot) - \mathbb{E}_{s',a' \sim \phi^{t-1}} r_h(s', a', \cdot, \cdot) \right\rangle + \mathbb{D}(\psi, \psi_h^t)$$

(Renaming the optimization variable and $\arg\max_x f(x) = \arg\min_x -f(x)$)

$$= \psi_h^{t+1}.$$

$\square$

## E IMPLEMENTATION OF ALGORITHM 3 WITH UPDATES OVER POLICIES.

In this section, we explain how the update in Algorithm 3 for different choices of $\mathbb{D}$. In both cases, we will derive an update that can be summarized by following template. Let us define $r_h^t(s, a) = \mathbb{E}_{s',a' \sim d_h^t} r(s, a, s', a')$ and $r_h^{t-1}(s, a) = \mathbb{E}_{s',a' \sim d_h^{t-1}} r(s, a, s', a')$

- Compute the $Q_h^t$ function corresponding to the reward function $2r_h^t - r_h^{t-1}$ minimizing a loss function that depends on the choice of $\mathbb{D}$.
- Update the policy as

$$\pi_h^{t+1}(a|s) \propto \pi_h^t(a|s) \exp\left(\beta Q_h^t(s, a)\right).$$

Finally, in Appx. E.3 we show that for $\mathbb{D}$ being the conditional relative entropy and for $\beta$ small enough the value function $Q_h^t$ is well approximated by the standard Bellman equations.

**Remark 6.** *Both choices of the Bregman divergence are 1 strongly convex so Thm. 5 applies with $\lambda = 1$.*

In the following we consider a generic reward function $\tilde{r}$. In our setting, we will apply the following results for $\tilde{r}_h = 2r_h^t - r_h^{t-1}$ in order to implement the updates of Alg. 3 for the different values of $h$ and $t$.

### E.1 $\mathbb{D}$ CHOSEN AS THE SUM OF CONDITIONAL AND RELATIVE ENTROPY

In this section, we explain how to implement the occupancy measure update in Algorithm 3 over policies. We use the machinery for single agent MDPs introduced in Bas-Serrano et al. (2021). In particular, we consider the Bregman divergence given by the sum of the relative entropy $D(d, d') = \sum_{s,a} d(s, a) \log \left( \frac{d(s,a)}{d'(s,a)} \right)$ and of the conditional relative entropy given, i.e. $H(d, d') = \sum_{s,a} d(s, a) \log \left( \frac{\pi_d(a|s)}{\pi_{d'}(a|s)} \right)$ with $\pi_d(a|s) = d(s, a) / \sum_a d(s, a)$. Under this choice for $\mathbb{D}$, the update of Algorithm 3 for particular values of $h, t, s_1$ corresponds to the solution of the following optimization program

$$d_h^{t+1} = \arg\max_{d \in \Delta^H} \sum_{h=1}^{H} \langle d_h, \tilde{r}_h \rangle - \frac{1}{\beta} D(d_h, d_h^t) - \frac{1}{\beta} H(d_h, d_h^t),$$

$$\text{s.t.} \quad E^T d_h = F^T d_{h-1} \quad \forall h \in [H]. \tag{Update I}$$

**Theorem 7.** *The policy $\pi_h^{t+1}$ with occupancy measure $d_h^{t+1}$ defined in Eq. (Update I) can be computed as follows*

$$\pi_h^{t+1}(a|s) \propto \pi_h^t(a|s) \exp\left(\beta Q_h^t(s, a)\right),$$

*where $Q_h^t$ is the minimizer of the following loss*

$$\frac{1}{\beta} \sum_{h=1}^{H} \log \sum_{s,a} \mu_h^t(s, a) \exp\left(\beta(2\tilde{r}_h + PV_{h+1} - Q_h)(s, a)\right) + \langle \nu_1, V_1 \rangle,$$

*while $V_{h+1}^t$ is given by the following closed form.*

$$V_{h+1}^t(s) = \frac{1}{\beta} \log \sum_{a} \pi_h^t(a|s) \exp(\beta Q_{h+1}^t(s, a)).$$

*Proof.* Let us introduce an auxiliary variable $\mu_h = d_h$ for all $h \in [H]$, then we can rewrite the optimization program as

$$\arg\max_{d \in \Delta^H} \max_{\mu \in \Delta^H} \sum_{h=1}^{H} \langle \mu_h, \tilde{r}_h \rangle - \frac{1}{\beta} D(\mu_h, \mu_h^t) - \frac{1}{\beta} H(d_h, d_h^t),$$

$$\text{s.t.} \quad E^T d_h = F^T \mu_{h-1} \quad \forall h \in [H],$$

$$\text{s.t.} \quad \mu_h = d_h \quad \forall h \in [H].$$

Then, by Lagrangian duality we have that

$$\max_{d \in \Delta^H} \max_{\mu \in \Delta^H} \min_{Q,V} \sum_{h=1}^{H} \langle \mu_h, \tilde{r} \rangle - \frac{1}{\beta} D(\mu_h, \mu_h^t) - \frac{1}{\beta} H(d_h, d_h^t)$$

$$+ \left\langle -E^T d_h + F^T \mu_{h-1}, V_h \right\rangle + \langle Q_h, d_h - \mu_h \rangle$$

$$= \max_{d \in \Delta^H} \max_{\mu \in \Delta^H} \min_{Q,V} \sum_{h=1}^{H} \langle \mu_h, \tilde{r} + FV_{h+1} - Q_h \rangle + \langle d_h, Q_h - EV_h \rangle$$

$$- \frac{1}{\beta} D(\mu_h, \mu_h^t) - \frac{1}{\beta} H(d_h, d_h^t)$$

$$+ \langle \nu_1, V_1 \rangle = \mathcal{L}^\star.$$

Then, by Lagrangian duality, we have that the objective is unchanged by swapping the min and max

$$\mathcal{L}^\star = \min_{Q,V} \max_{d \in \Delta^H} \max_{\mu \in \Delta^H} \sum_{h=1}^{H} \langle \mu_h, \tilde{r}_h + FV_{h+1} - Q_h \rangle + \langle d_h, Q_h - EV_h \rangle$$

$$- \frac{1}{\beta} D(\mu_h, \mu_h^t) - \frac{1}{\beta} H(d_h, d_h^t) + \langle \nu_1, V_1 \rangle.$$

The inner maximization is solved by the following values

$$\mu_h^+(Q, V) \propto \mu_h^t \odot \exp\left(\beta(\tilde{r}_h + FV_{h+1} - Q_h)\right),$$
$$\pi_h^+(Q, V; s) \propto \pi_h^t(\cdot|s) \odot \exp\left(\beta(Q_h(s, \cdot) - V_h(s))\right),$$

where $\odot$ denotes the elementwise product between vectors. Then, replacing these values in the Lagrandian and parameterizing the functions $V_h$ by the functions $Q_h$ to ensure normalization of the policy, i.e. $V_h(s) = \frac{1}{\beta} \log \sum_a \pi_h^t(a|s) \exp(\beta Q_h(s, a))$ we have that

$$\mathcal{L}^\star = \min_Q \frac{1}{\beta} \sum_{h=1}^{H} \log \sum_{s,a} \mu_h^t(s, a) \exp\left(\beta(\tilde{r}_h + FV_{h+1} - Q_h)(s, a)\right) + \langle \nu_1, V_1 \rangle.$$

Therefore, denoting

$$Q_h^t = \arg\min_Q \frac{1}{\beta} \sum_{h=1}^{H} \log \sum_{s,a} \mu_h^t(s, a) \exp\left(\beta(\tilde{r}_h + FV_{h+1} - Q_h)(s, a)\right) + \langle \nu_1, V_1 \rangle,$$

and $V_h^t = \frac{1}{\beta} \log \sum_a \pi_h^t(a|s) \exp(\beta Q_h^t(s, a))$, we have that the policy $\pi_h^{t+1}(\cdot|s) = \pi_h^+(Q^t, V^t; s)$ has occupancy measure equal to $d_h^{t+1}$ for all $h \in [H]$. This is because by the constraints of the problem we have that $d_h^{t+1}$ satisfies the Bellman flow constraints and that the policy $\pi_h^{t+1}$ satisfies $\pi_h^{t+1}(a|s) = d_h^t(s, a)/\sum_a d_h^t(s, a)$. $\qquad\square$

### E.2 $\mathbb{D}$ CHOSEN AS CONDITIONAL RELATIVE ENTROPY NEU ET AL. (2017)

In this section, we study the update considering $\mathbb{D}$ chosen as sum of the conditional relative entropy over the stages $h'$ s.t. $1 \le h' \le h$, i.e. we study the following update.[5]

$$d^{t+1} = \arg\max_{d \in \Delta^H} \sum_{h=1}^{H} \left( \langle d_h, \tilde{r}_h \rangle - \frac{1}{\beta} \sum_{h'=1}^{h} H(d_{h'}, d_{h'}^t) \right),$$
$$\text{s.t.} \quad E^T d_h = F^T d_{h-1} \quad \forall h \in [H]. \tag{6}$$

**Theorem 8.** *The policy $\pi_h^{t+1}$ with occupancy measure $d_h^{t+1}$ defined in Eq. (6) can be computed as follows*

$$\pi_h^{t+1}(a|s) \propto \pi_h^t(a|s) \exp\left( \frac{\beta}{H - h + 1}(Q_h^t(s, a)) \right),$$

*where $Q_h^t$ and $V_{h+1}^t$ satisfies the following recursion*

$$Q_h^t = \tilde{r}_h + FV_{h+1}^t$$
$$V_{h+1}^t(s) = \frac{H - h + 1}{\beta} \log \sum_a \pi_h^t(a|s) \exp\left( \frac{\beta}{H - h + 1} Q_{h+1}^t(s, a) \right).$$

**Remark 7.** *The above recurrencies are sometimes called soft Bellman equations Ziebart (2010); Fox et al. (2015).*

*Proof.* Let us introduce an auxiliary variable $\mu_h = d_h$ for all $h \in [H]$, then we can rewrite the optimization program as

$$\arg\max_{d \in \Delta^H} \max_{\mu} \sum_{h=1}^{H} \left( \langle \mu_h, \tilde{r}_h \rangle - \frac{1}{\beta} \sum_{h'=1}^{h} H(d_{h'}, d_{h'}^t) \right)$$
$$\text{s.t.} \quad E^T d_h = F^T \mu_{h-1} \quad \forall h \in [H]$$
$$\text{s.t.} \quad \mu_h = d_h \quad \forall h \in [H].$$

---

[5]The sum over previous stages is taken to ensure 1-strong convexity. Indeed, it holds that $\sum_{h'=1}^{h} H(d_{h'}, d_{h'}^t) \ge D(d_h, d_h') \ge \frac{1}{2}\|d_h - d_h'\|_1^2$. The first inequality is proven in (Neu & Olkhovskaya, 2021, Lemma 7).

Notice that importantly, we do not constrain the variable $\mu$. Then, by Lagrangian duality we have that

$$\max_{d \in \Delta^H} \max_{\mu} \min_{Q,V} \sum_{h=1}^{H} \langle \mu_h, \tilde{r}_h \rangle - \frac{1}{\beta} \sum_{h'=1}^{h} H(d_{h'}, d_{h'}^t)$$

$$+ \langle -E^T d_h + F^T \mu_{h-1}, V_h \rangle + \langle Q_h, d_h - \mu_h \rangle$$

$$= \max_{d \in \Delta^H} \max_{\mu} \min_{Q,V} \sum_{h=1}^{H} \langle \mu_h, \tilde{r}_h + FV_{h+1} - Q_h \rangle + \langle d_h, Q_h - EV_h \rangle$$

$$- \frac{1}{\beta} \sum_{h'=1}^{h} H(d_{h'}, d_{h'}^t) + \langle \nu_1, V_1 \rangle$$

$$= \min_{Q,V} \max_{d \in \Delta^H} \max_{\mu} \sum_{h=1}^{H} \langle \mu_h, \tilde{r}_h + FV_{h+1} - Q_h \rangle + \langle d_h, Q_h - EV_h \rangle$$

$$- \frac{H-h+1}{\beta} H(d_h, d_h^t) + \langle \nu_1, V_1 \rangle = \tilde{\mathcal{L}}^\star,$$

where the last equality holds by Lagrangian duality and by $\sum_{h=1}^{H} \sum_{h'=1}^{h} H(d_{h'}, d_{h'}^t) = \sum_{h=1}^{H}(H-h+1)H(d_{h'}, d_{h'}^t)$. Now since $\mu$ is unconstrained we have that $\max_{\mu} \sum_{h=1}^{H} \langle \mu_h, \tilde{r}_h + FV_{h+1} - Q_h \rangle$ is equivalent to impose the constraint $\tilde{r}_h + FV_{h+1} = Q_h$ for all $h \in [H]$. Moreover, as in the proof of Thm. 7 the optimal $d_h$ needs to satisfies that $\pi_{d_h}(a|s) = d_h(s,a)/\sum_a d_h(s,a)$ is equal to $\pi_h^+(Q,V;s) = \pi_h^t(\cdot|s) \odot \exp\left(\frac{\beta}{H-h+1}(Q_h(s,\cdot) - V_h(s))\right)$ for $V_h(s) = \frac{H-h+1}{\beta} \log \sum_a \pi_h^t(a|s) \exp(\frac{\beta}{H-h+1} Q_h(s,a))$. Plugging in, these facts in the expression for $\tilde{\mathcal{L}}^\star$, we have that

$$\tilde{\mathcal{L}}^\star = \min_{Q} \langle \nu_1, V_1 \rangle \quad \text{s.t.} \quad \tilde{r}_h + FV_{h+1} = Q_h \quad \forall h \in [H].$$

Since the above problem as only one feasible point, we have that the solution is the sequence $Q_h^t$ satisfying the recursion $\tilde{r}_h + FV_{h+1}^t = Q_h^t$ with $V_h^t(s) = \frac{H-h+1}{\beta} \log \sum_a \pi_h^t(a|s) \exp(\frac{\beta}{H-h+1} Q_h^t(s,a))$. $\qquad \square$

### E.3 Approximating soft Bellman equations by standard Bellman equations.

Unfortunately, implementing the update for the $V$ value as in Theorem 7 is often numerically instable. In this section, we show a practical approximation which is easy to implement and shown to be accurate for $\beta$ sufficiently small.

**Theorem 9.** *Let us denote $\beta_h = \frac{\beta}{H-h+1}$ and let us assume that the values $Q_h^t$ generated by the soft Bellman equations in Thm. 8 are uniformly upper bounded by $Q_{\max}$, and let us choose $\beta_h \leq \frac{1}{Q_{\max}}$ for all $h \in [H]$. Then, it holds that*

$$\langle \pi_h^t(\cdot|s), Q_h^t(s,\cdot) \rangle \leq \frac{1}{\beta_h} \log \sum_a \pi_h^t(a|s) \exp(\beta_h Q_h^t(s,a)) \leq \langle \pi_h^t(\cdot|s), Q_h^t(s,\cdot) \rangle + \beta_h Q_{\max}^2.$$

*Proof.*

$$\frac{1}{\beta_h} \log \sum_a \pi_h^t(a|s) \exp(\beta_h Q_h^t(s,a)) \geq \frac{1}{\beta_h} \sum_a \pi_h^t(a|s) \log \exp(\beta_h Q_h^t(s,a))$$

$$= \langle \pi_h^t(\cdot|s), Q_h^t(s,\cdot) \rangle,$$

where the above inequality holds for Jensen's. For the upper bound, we first use the inequality $e^x \leq 1 + x + x^2$ for $x \leq 1$ we have that

$$\frac{1}{\beta_h} \log \sum_a \pi_h^t \exp(\beta_h Q_h^t(s,a))$$

$$\leq \frac{1}{\beta_h} \log \sum_a \pi_h^t (1 + \beta_h Q_h^t(s,a) + \beta_h^2 Q_{\max}^2) \quad \text{(Using } Q_h^t(s,a) \leq Q_{\max})$$

$$= \frac{1}{\beta_h} \log(1 + \beta_h \sum_a \pi_h^t(a|s) Q_h^t(s,a) + \beta_h^2 Q_{\max}^2)$$

$$\leq \frac{1}{\beta_h} \left( \sum_a \pi_h^t(a|s) \beta_h Q_h^t(s,a) + \beta_h^2 Q_{\max}^2 \right) \quad \text{(Using } \log(1+x) \leq x)$$

$$\leq \langle \pi_h^t(\cdot|s), Q_h^t(s,\cdot) \rangle + \beta_h Q_{\max}^2.$$

$\square$

**Remark 8.** *Given this result, in the implementation for deep RL experiment, i.e. Algorithm 4 we compute the standard Q value satisfying the standard Bellman equations (given in Lemma 1) rather than the soft Bellman equation in Thm. 7. In virtue of Thm. 9, the approximation is good for $\beta$ reasonably small.*

# F ADDITIONAL EXPERIMENT

## F.1 EXPERIMENT IN MT-BENCH 101

The tasks in MT-bench 101 include Context Memory (CM), Anaphora Resolution (AR), Separate Input (SI), Topic Shift (TS), Content Confusion (CC), Content Rephrasing (CR), Format Rephrasing (FR), Self-correction (SC), Self-affirmation (SA), Mathematical Reasoning (MR), General Reasoning (GR), Instruction Clarification (IC), and Proactive Interaction (PI). We list the description of each task in Tab. 3. The default evaluation mode of MT-bench 101 is that the GPT model requires to access the conversation based on the given ground truth of previous steps, provided in MT-bench 101. However, in our problem setting, the answers among the conversation is also generated by the model. We use "gpt-4o-mini-2024-07-18" to evaluate the conversation. The maximum output length and maximum sequence length of gpt-4o are set as 4096. We use a batch size of 8 with a temperature of 0.8. We use the same prompt for gpt-4o as in Bai et al. (2024). Our experiment is conducted on 4 H200 GPUs. We use the PyTorch platform and the Transformer Reinforcement Learning (TRL) for finetuning.

Table 3: A detailed description of each task in MT-bench 101 (taken from Bai et al. (2024).)

| Task | Abbr. | Description |
|------|-------|-------------|
| Context Memory | CM | Recall early dialogue details to address the user's current question. |
| Anaphora Resolution | AR | Identify pronoun referents throughout a multi-turn dialogue. |
| Separate Input | SI | The first turn outlines the task requirements and the following turns specify the task input. |
| Topic Shift | TS | Recognize and focus on the new topic when users unpredictably switch topics. |
| Content Confusion | CC | Avoid interference from similar-looking queries with distinct meanings in the dialogue's history. |
| Content Rephrasing | CR | Rephrase the content of the last response according to the user's newest requirement. |
| Format Rephrasing | FR | Rephrase the format of the last response according to the user's newest requirement. |
| Self-correction | SC | Recorrect the last response according to the user feedback. |
| Self-affirmation | SA | Preserve the last response against inaccurate user feedback. |
| Mathematical Reasoning | MR | Collaboratively solve complex mathematical problems with users across dialogue turns. |
| General Reasoning | GR | Collaboratively solve complex general reasoning problems with users across dialogue turns. |
| Instruction Clarification | IC | Seek clarification by asking further questions on ambiguous user queries. |
| Proactive Interaction | PI | Propose questions in reaction to user statements to spark their interest to continue the dialogue. |

Next, we provide the comparison between the proposed `MPO` and IPO (Azar et al., 2024), which also uses the squared loss and bypasses the BT model assumption. We run both IPO and MPO for one iteration. The results in Tab. 4 show that `MPO` achieves a higher average score than IPO.

Table 4: Comparison between `MPO` and IPO in MT-BENCH 101 dataset.

| Model | Avg. | Perceptivity | | | | | Adaptability | | | | | | Interactivity | |
| | | Memory | Understanding | | Interference | | Rephrasing | | Reflection | | Reasoning | | Questioning | |
| | | CM | SI | AR | TS | CC | CR | FR | SC | SA | MR | GR | IC | PI |
| Base (Mistral-7B-Instruct) | 6.223 | 7.202 | 7.141 | 7.477 | 7.839 | 8.294 | 6.526 | 6.480 | 4.123 | 4.836 | 4.455 | 5.061 | 5.818 | 5.641 |
| IPO | 6.498 | 7.518 | 7.480 | 7.759 | 7.952 | 8.652 | 6.892 | 6.768 | 4.390 | 5.185 | 4.313 | 5.378 | 6.146 | 6.044 |
| MPO | 6.630 | 7.624 | 7.846 | 8.085 | 8.398 | 8.947 | 7.105 | 7.286 | 4.208 | 4.993 | 4.377 | 5.264 | 6.179 | 5.873 |

We now present an ablation study to evaluate the benefits of incorporating terminal rewards. Using MPO, we compare two approaches for optimizing $a_h$: one computes the preference signal based on the terminal state $s_{H+1}$, while the other uses the immediate next state $s_h$. The results within one iteration for the MT-Bench 101 dataset are shown in Tab. 5, and those for the GSM/Math experiments are provided in Tab. 6. Our findings reveal that using the terminal state $s_{H+1}$ performs worse than using the immediate state $s_h$ in MT-Bench 101. In contrast, the difference in performance is negligible in the GSM/Math tasks. The underlying reason is that in multi-turn conversational datasets, especially when adjacent questions are not closely related, relying on preferences derived from the terminal state can introduce noise. However, in math and reasoning tasks, the terminal state often captures the final answer, making it more critical. Moreover, using $s_{H+1}$ for preference signals is significantly more computationally expensive than using $s_h$, due to the extended sequence length. Consequently, we conclude that adapting the choice of terminal preference or intermediate preference on the task's characteristics is crucial for balancing performance and efficiency.

## F.2 TABULAR EXPERIMENT

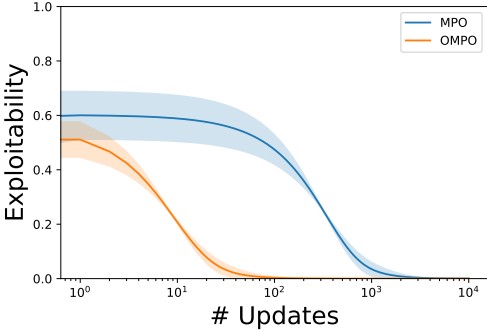

Figure 2: Results in the tabular experiments. Curves are averages across 10 different randomly generated environments. The error bars report one standard deviation.

The setting of our large-scale experiments does not match the assumptions under which Thm. 5 is proven. In particular, in the large scale experiments the state action value functions can not be computed exactly. In this section, we consider a synthetic experiment in which the state action functions can be computed exactly for both `OMPO` and `MPO`. We generate 10 random gridworlds with a number of states and actions sample uniformly from the intervals $[1, 100]$ and $[2, 10]$. We plot the exploitability computed as

$$\left\langle \nu_1, \max_{\pi} V^{\pi, \pi^k} - V^{\pi^k \pi^k} \right\rangle$$

which is a standard metric to evaluate the distance from a Nash equilibrium. In particular, when $(\pi^k, \pi^k)$ is a Nash equilibrium, the exploitability is 0. We can see that `OMPO` achieves very low exploitability after 100 updates while 2000 updates are needed by `MPO`. In this case, where the

Table 5: Ablation on terminal reward in MT-BENCH 101 dataset.

| Model | Avg. | Perceptivity | | | | | Adaptability | | | | | | Interactivity | |
| | | Memory | Understanding | | Interference | | Rephrasing | | Reflection | | Reasoning | | Questioning | |
| | | CM | SI | AR | TS | CC | CR | FR | SC | SA | MR | GR | IC | PI |
|---|---|---|---|---|---|---|---|---|---|---|---|---|---|---|
| Base (Mistral-7B-Instruct) | 6.223 | 7.202 | 7.141 | 7.477 | 7.839 | 8.294 | 6.526 | 6.480 | 4.123 | 4.836 | 4.455 | 5.061 | 5.818 | 5.641 |
| MPO (intermediate reward) | 6.630 | 7.624 | 7.846 | 8.085 | 8.398 | 8.947 | 7.105 | 7.286 | 4.208 | 4.993 | 4.377 | 5.264 | 6.179 | 5.873 |
| MPO (terminal reward) | 6.459 | 7.536 | 7.328 | 7.643 | 8.084 | 8.518 | 6.847 | 6.883 | 4.357 | 4.863 | 4.403 | 5.542 | 6.034 | 5.924 |

Table 6: Ablation on terminal reward in MATH and GSM8K dataset.

| Method | GSM8K | Math |
|---|---|---|
| Base (Qwen2-7B-Instruct) | 0.8559 | 0.5538 |
| MPO (intermediate reward) | 0.8734 | 0.5720 |
| MPO (terminal reward) | 0.8734 | 0.5734 |

$Q$ functions can be computed exactly, we can appreciate the faster convergence rate of OMPO as described by Thm. 5.

## F.3 EXPERIMENT ON MATH REASONING TASKS

As discussed in Appx. B, our framework can also cover the alignment of chain-of-thought reasoning. In this section, we validate the proposed methods on math reasoning tasks. We select two widely used datasets: MATH Hendrycks et al. (2021) and GSM8K Cobbe et al. (2021). We use Qwen2-7B-Instruct as the base model and follow the same evaluation procedure as in Lai et al. (2024). We adopt the dataset for alignment from Lai et al. (2024), which contains 10795 samples of augmented mathematical problems from MetaMath (Yu et al., 2024) and MMIQC (Liu et al., 2024b)[6]. For step-DPO, we use the checkpoint provided in Lai et al. (2024). For both MPO and OMPO, we perform full-parameter finetuning for 1 epoch with learning rate $5e^{-7}$ and $\beta$ tuned in the range of $\{0.1, 0.01, 0.001\}$. For both MPO and OMPO, we select the Llama-3-based model as the preference oracle[7] and set the $\log z$ are set as 0.5. The final state with the answer is important in this task so we only use the terminal reward (see Tab. 6 for comparison). We use AdamW optimizer (Loshchilov & Hutter, 2019) and cosine learning rate schedule (Loshchilov & Hutter, 2017) with a warmup ratio of 0.1. The experiment is conducted on 4 A100-SXM4-80GB GPUs. The result is provided in Tab. 7, showing that the proposed methods achieve performance comparable to step-DPO (Lai et al., 2024). Notably, MPO and OMPO do not require the ground truth label of the dataset during finetuning while Lai et al. (2024) requires it. Additionally, MPO and OMPO need only a Llama3-based pair-preference-model to compare two answers. Step-DPO requires GPT-4 to identify the incorrect reasoning step in an answer, which is a considerably more difficult task than comparison.

---

[6]https://huggingface.co/datasets/xinlai/Math-Step-DPO-10K
[7]https://huggingface.co/RLHFlow/pair-preference-model-LLaMA3-8B

Table 7: Performance of math reasoning on MATH and GSM8K dataset across various models. `MPO` and `OMPO` achieve comparable performance comparable to step-DPO without requiring the ground truth label of the dataset during fine-tuning while Lai et al. (2024) requires. Additionally, `MPO` and `OMPO` only need access to an oracle Llama-3 to compare two answers whereas step-DPO Lai et al. (2024) requires GPT-4 to locate the identify the incorrect reasoning step in an answer, which is a considerably more difficult task than comparison.

| Method | GSM8K | Math |
|---|---|---|
| Base (Qwen2-7B-Instruct) | 0.8559 | 0.5538 |
| Step-DPO (Lai et al., 2024) | 0.8680 | **0.5836** |
| `MPO` (iter=1) | 0.8734 | 0.5734 |
| `MPO` (iter=2) | 0.8734 | 0.5786 |
| `OMPO` (iter=2) | **0.8779** | 0.5786 |

## G  MOTIVATION OF CONSIDERING INTERMEDIATE REWARD

In this section, we elaborate on the motivation for considering intermediate rewards at each turn instead of only terminal rewards.

In multi-turn conversation tasks, such as MT-bench 101 (Bai et al., 2024), the user asks questions $x_1, x_2, x_3$, and receives answers $a_1, a_2, a_3$. When $x_2$ is not closely related to $x_1$, aligning the first step using feedback among different $a_1$ is much more helpful than using the sequence $[a_1, x_2, a_2]$, where $x_2, a_2$ can be considered as noise.

In mathematical reasoning tasks, as mentioned in Lai et al. (2024), some cases yield correct final answers but contain errors in intermediate reasoning steps. Consequently, Lai et al. (2024) filter out such samples using GPT-4. For example, consider a case where the reasoning steps yield a correct final answer but include an error: $[a_1^{\text{correct}}, a_2^{\text{wrong}}, a_3^{\text{correct}}]$, where $a_2^{\text{wrong}}$ is incorrect while all of the other steps and the final answer $a_3^{\text{correct}}$ is correct. When there is another response, $[a_1^{\text{correct}}, a_2^{\text{correct}}, a_3^{\text{correct}}]$ with all correct steps, using only terminal signal for aligning step 2 might not guarantee that $a_2^{\text{correct}} \succ a_2^{\text{wrong}}$ because both of final answers are correct, especially when there is only an incorrect step among long reasoning steps. In contrast, an intermediate signal would clearly indicate $a_2^{\text{correct}} \succ a_2^{\text{wrong}}$, accurately reflecting the quality of the intermediate steps. In practice, if the final signal is important, e.g., in math reasoning task, then we can use only the terminal reward or the average of terminal reward and intermediate reward, otherwise one can just use the intermediate reward, which is cheaper to collect as compared to assigning reward until the terminal state.

