# OpenReview forum: "Multi-Step Preference Optimization via Two-Player Markov Games"
_ICLR.cc/2025/Conference — Submitted to ICLR 2025_

### Official Review · Reviewer_896K · 2024-10-27

**Soundness:** 2
**Presentation:** 3
**Contribution:** 2
**Rating:** 6
**Confidence:** 4

**Summary:**

This work addresses the problem of multi-step preference optimization using a two-player partially observable Markov game. The authors introduce Multi-step Preference Optimization (MPO) based on the natural actor-critic framework and Optimistic Multi-step Preference Optimization (0MPO), built upon the optimistic online gradient descent.  Experiment results show that the proposed methods achieve considerable improvement on multi-turn conversation datasets, e.g., MT-bench-101, compared to the multi-step variant of DPO, e.g., iterative SPPO, iterative step-DPO.

**Strengths:**

1. The paper writing is very clear and easy to follow.
2. The theory is sound and solid.

**Weaknesses:**

1. The multi-step preference optimization problem is discussed in other contexts such as token-level single-step alignment. It would be beneficial to incorporate some discussions with them in the paper.

    1. From r to Q*: Your Language Model is Secretly a Q-Function, Rafailov et al., COLM 2024
    2. Token-level Token-level Direct Preference Optimization, Zeng et al., ICML 2024
    3. TIS-DPO: Token-level Importance Sampling for Direct Preference Optimization With Estimated Weights, Liu et al.

2. Could the authors elaborate on the distinction between this work and the extension of the SPPO's framework to a multi-turn setup? The latter is also a self-play-based alignment method that perceives the issue as a constant-sum, two-player game with the goal of identifying the Nash equilibrium policy.

    1. Self-Play Preference Optimization for Language Model Alignment, Wu et al.


3. Current empirical improvement from MPO/OMPO over baselines like Step-DPO seems not so obvious. The authors may consider including more experiments more than just multi-turn conversations in MT-bench.

**Questions:**

Please see above

---

### Official Review · Reviewer_DzfW · 2024-11-03

**Soundness:** 2
**Presentation:** 2
**Contribution:** 1
**Rating:** 3
**Confidence:** 3

**Summary:**

This paper studies the multi-turn preference-based setting and applies it to large language model training. They frame multi-step RLHF as a two-player Markov game and propose algorithms MPO/OMPO.

**Strengths:**

The paper frames the problem of multi-step RLHF as a two-play Markov game. It proposes and studies two algorithms that over-perform their one-turn counter parts according to the experiments.

**Weaknesses:**

The authors propose an idea that is pretty close to other existing works in the literature. The main difference is that the signal used for optimization, which is the preference, is turn/state-based instead of a final preference. The authors need to better motivate why this choice is important as this is the only thing that differentiate them from the past literature and present concrete scenarios where this data can be collected from humans. Previous works have argued that they specifically choose the final preference case because human data collection is expensive and complex when humans have to provide a preference signal at each turn. Have the authors realized an experimental data collection involving humans and turn-based preference collection?

The authors present a lot of different results but most of them should already exist in the literature of online mirror descent (exemple theorem 4). I think the authors should make it clearer that those results are derived from previous ones and frame them as such in the main text. Lemma 1 is a known result and Lemma 2 should have existing variants. The derivation of the practical algorithm is also highly inspired by the IPO derivation where the Q function replaces the one-step preference (step 3-5). This should be clearly stated in the text.

The experiments need a multi-step baseline where the Q function only uses the preference at the last step to see if it is useful to have an intermediate signal. Not sure if this is the result presented in figure 1b, it was not really clear in the text. If this is the case it seems that it is not really useful to have a signal at each turn. In addition, more details on the one-step baselines such as DPO should be provided. Especially if there is masking of the user turns for DPO and why not compare it  to IPO as it seems you are using a square loss. It would make more sense to use a one-step algorithm using the same type of loss.

I am willing to improve my score if:
 - The authors reframe their theoretical results by stating clearly in the main text that they derive their results from known proofs.
 - The authors provide a clear motivation of the turn-based preference signal for the LLM case and how they can build a data-collection involving humans where collecting such preferences makes sense.
 - The authors provide better baselines for their experiment such as MPO with end-signal only versus MPO with intermediary signal. Use IPO instead of DPO as it uses a squared-loss which is closer to MPO.

**Questions:**

See section weaknesses.

---

### Official Review · Reviewer_FmdH · 2024-11-03

**Soundness:** 3
**Presentation:** 3
**Contribution:** 3
**Rating:** 6
**Confidence:** 3

**Summary:**

The paper studies multi-step RLHF which formulates the alignment into a multi-step two-player constant-sum Markov game rather than bandits. The authors propose Multi-Step Preference Optimization (MPO) based on the natural actor-critic framework, and OMPO which utilizes optimistic online gradient descent by considering the occupancy measure space. Theoretical analysis demonstrates that OMPO has a convergence rate $1/\epsilon$ to an approximate Nash equilibrium, which is more efficient compared to existing methods. Experiments on the MT-bench-101 dataset show that both MPO and OMPO outperform traditional methods in multi-turn conversational tasks with respect to certain scores.

**Strengths:**

1. Formulating RLHF as a two-player zero-sum Markov games is a convincing approach because there are many existing works that use normal games to formulate RLHF in single-turn tasks.
2. The OMPO algorithm is novel and the derivation is not trivial. The theoretical guarantee is valid and the faster convergence rate is promising.
3. The empirical performance of MPO and OMPO seems better than multi-step DPO and SPPO, which is not surprising due to the multi-turn nature of the tasks.

**Weaknesses:**

1. The theoretical analysis assumes one can attain the occupancy measure and reward function accurately, which is unrealistic in practice. It would be more convincing to consider online estimation error and sample complexity.
2. Although OMPO has a faster theoretical convergence rate, the empirical performance between MPO and OMPO seems close. More discussion about this would be meaningful, which will also shed light on how to further improve OMPO.

**Questions:**

Why is the algorithm called 0MPO in the paper? It seems that OMPO makes more sense due to 'Optimistic'.

---

### Official Review · Reviewer_Lfse · 2024-11-04

**Soundness:** 2
**Presentation:** 3
**Contribution:** 3
**Rating:** 6
**Confidence:** 3

**Summary:**

This paper introduces Multi-step Preference Optimization (MPO) and Optimistic MPO (0MPO) for aligning large language models with human preferences in multi-turn conversations. By modeling the problem as a two-player Markov game, these algorithms aim to maximize player's winning rate across conversation steps. Theoretical analysis shows 0MPO converges to an $$\epsilon$$-approximate Nash equilibrium in $$O(\epsilon^{-1})$$ updates. And experiments on the MT-bench-101 dataset demonstrate MPO and 0MPO's significant improvements over baselines in perceptivity, adaptability, and interactivity.

**Strengths:**

1. This paper introduces a new approach by modeling multi-step preference optimization as a two-player Markov game, which is interesting and more suitable for real-world multi-turn conversations.
2. In addition, the authors further provide a rigorous theoretical analysis, showing the proposed 0MPO converges to an $$\epsilon$$-approximate Nash equilibrium in $$O(\epsilon^{-1})$$ policy updates, which is more efficient than previous algorithms.

**Weaknesses:**

1. The primary weakness of this paper is noted by the authors, where the fixed and finite-horizon assumption in their framework may not be suitable for real-world conversation or reasoning process, since their steps usually differ in various situations.
2. The experimental results are primarily validated on the MT-bench-101 dataset and a few baselines. Further experiments on diverse datasets and other base models may be necessary to establish robustness and generalizability of the proposed methods.
3. The empirical validation relies on automatic metrics evaluated by gpt-4o. The reliance on it leaves room for uncertainty about how well the model captures nuanced human preferences in real-world scenarios. It would be better to have human evaluations on different methods when facing the same situations.

**Questions:**

1. Since the authors use the Monte Carlo estimator in Eq. (5), how many rollouts are required at the minimum to well estimate the Q value? Or in other words, how much impact does the number of samples have on the performance of the proposed methods?
2. Did the authors consider applying the MPO and 0MPO methods to other base models besides Mistral-7B-Instruct, like llama, Qwen, etc.?
3. As the methods proposed in this paper are for multi-step RLHF, is the evaluation of experimental results conducted on the entire conversation or in each conversation turn?
4. Some typos:
  1. It seems that there is a problem with the citation format in Line 48-49.
  2. The usage of preference oracle $$o(\cdot)$$ in Line 144 is different from the definition in Line 142.

---

### Meta-Review · Area_Chair_UiH7 · 2024-12-21

**Metareview:**

The paper presents multi steps preference optimization, an actor critic framework to tackle conversations as a two-player constant-sum Markov game.
There are few weaknesses, including the finite horizon assumption and limited experiments, and the relative similarity with other game-theoretic approaches to alignment.
The experiments could also be more thorough, and there is space for additional comparison—see reviewers detailed comments

**Additional Comments On Reviewer Discussion:**

One reviewer did not fully engage on the discussion.

---

### Decision · Program_Chairs · 2025-01-22

Reject